# Single cell transcriptomes and multiscale networks from persons with and without Alzheimer's disease

Qi Wang [1], Jerry Antone[2], Eric Alsop [2], Rebecca Reiman[2], Cory Funk [3], Jaroslav Bendl [4], Joel T. Dudley[1], Winnie S. Liang[2], Timothy L. Karr [1], Panos Roussos [4], David A. Bennett[5], Philip L. De Jager [6], Geidy E. Serrano[7], Thomas G. Beach[7], Kendall Van Keuren-Jensen[2], Diego Mastroeni [1], Eric M. Reiman [8,9] & Benjamin P. Readhead [1,9] ✉

The emergence of single nucleus RNA sequencing (snRNA-seq) offers to revolutionize the study of Alzheimer's disease (AD). Integration with complementary multiomics data such as genetics, proteomics and clinical data provides powerful opportunities to link cell subpopulations and molecular networks with a broader disease-relevant context. We report snRNA-seq profiles from superior frontal gyrus samples from 101 well characterized subjects from the Banner Brain and Body Donation Program in combination with whole genome sequences. We report findings that link common AD risk variants with *CR1* expression in oligodendrocytes as well as alterations in hematological parameters. We observed an AD-associated CD83(+) microglial subtype with unique molecular networks and which is associated with immunoglobulin IgG4 production in the transverse colon. Our major observations were replicated in two additional, independent snRNA-seq data sets. These findings illustrate the power of multi-tissue molecular profiling to contextualize snRNA-seq brain transcriptomics and reveal disease biology.

Single-cell sequencing technologies such as snRNA-seq, in combination with the steady development of analytical methods, have greatly advanced our understanding of complex human diseases in the past decade[1]. These techniques have been employed to study Alzheimer's disease (AD), a devastating neurodegenerative disease characterized by the development of brain neuropathologies including neuritic plaques and neurofibrillary tangles leading to impaired cognition, with the goal of interpreting dynamic molecular processes within and across cell types[2–11]. Delineating cell type-specific changes and dysregulation in AD at the single cell level is crucial for deciphering the

molecular mechanisms underpinning the onset and progression of AD, thus enabling the discovery of novel drug targets and the development of effective therapeutic strategies[12].

Findings from large-scale genetic studies of AD risk have convincingly implicated microglial biology as a critical causal component of AD onset and progression, including important roles in amyloid clearance[13] and immune response in the presence of tau pathology[14]. These responses involve a specific transcriptional state referred to as activation response microglia (ARM)[15], disease-associated microglia (DAM)[16], or MicroGlial neuroDegenerative phenotype (MGnD)[17], which

[1]ASU-Banner Neurodegenerative Disease Research Center, Arizona State University, Tempe, AZ 85281, USA. [2]Division of Neurogenomics, The Translational Genomics Research Institute, Phoenix, AZ 85004, USA. [3]Institute for Systems Biology, Seattle, WA 98109, USA. [4]Department of Genetics and Genomic Sciences, Icahn School of Medicine at Mount Sinai, New York, NY 10029, USA. [5]Rush Alzheimer's Disease Center, Rush University Medical Center, Chicago, IL 60612, USA. [6]Department of Neurology, Center for Translational and Computational Neuroimmunology, Columbia University Irving Medical Center, New York, NY 10032, USA. [7]Civin Laboratory for Neuropathology, Banner Sun Health Research Institute, Sun City, AZ 85351, USA. [8]Banner Alzheimer's Institute, Phoenix, AZ 85006, USA. [9]These authors contributed equally: Eric M. Reiman, Benjamin P. Readhead ✉e-mail: ben.readhead@asu.edu

demonstrate activation signature genes that overlap considerably with AD risk genes identified in genome-wide association studies[18] (GWAS). These activation signatures, however, have not been fully captured in several recently reported snRNA-seq studies of microglia in frozen human AD postmortem brain tissues[8–10].

Despite the value of increasingly detailed molecular characterizations of brain tissue from subjects with AD, the development of a sophisticated understanding of the clinical and neuropathological context for identified cell subtypes and molecular networks is necessarily limited by the resolution of available antemortem and postmortem characterizations. Further, potentially informative cross-tissue interactions (e.g., gut–brain) are masked by a paucity of biorepositories that routinely collect brain and peripheral tissues from the same subjects. In addition to illuminating disease biology, multi-tissue profiling can offer valuable opportunities to identify peripheral biomarkers that might indicate disease-relevant brain states and treatment responses.

In this study, we generated snRNA-seq profiles from 481,840 nuclei collected from postmortem superior frontal gyrus (SFG) cortical tissue samples from 101 aged subjects with excellent clinical and postmortem neuropathological characterizations from the Arizona Study of Aging and Neurodegenerative Disorders/Brain and Body Donation Program (BBDP)[19]. By integrating whole genome sequencing (WGS) data, we report findings that link common AD risk variants with *CR1* expression in oligodendrocytes as well as alterations in peripheral hematocrit levels. We also applied multiscale network modeling approaches to learn the gene regulatory networks that characterize AD-associated cell subpopulations. Our findings have revealed a specific CD83(+) microglial subtype with unique molecular networks that encompass many known regulators of AD-relevant microglial biology, and which are associated with immunoglobulin production in the transverse colon. These findings demonstrate the power of multi-tissue molecular profiling to contextualize single-nucleus brain transcriptomics and thus illuminate disease biology. The transcriptomic, genetic, phenotypic, and network data resources described within this study are available for access and utilization by the scientific community.

## Results
### A public resource of single-cell transcriptome and other associated molecular data
We developed a shared resource of snRNA-seq data from SFG, along with WGS profiles, from very high-quality brain tissue (mean PMI = 3.4 h) from 101 well-characterized brain donors with and without the clinical and neuropathological features of AD. Detailed demographic, clinical, and postmortem neuropathological data from this cohort are reported in Table 1. We used the National Institute on Aging and Alzheimer's Association (NIA-AA) AD criteria[20] from neuropathological characterization to dichotomize the subjects, to facilitate targeted comparison in this study.

### Differentially abundant cell types in the Superior Frontal Gyrus of Alzheimer's disease
We performed Chromium 10× snRNA-seq on postmortem, SFG brain tissue samples from 101 donors (AD n = 66, aged controls n = 35). We applied a set of rigorous quality control criteria to exclude low-quality nuclei or doublets from each sample, then integrated all nuclei into a single data object (see Methods). After cell filtering, we retained a total of 481,840 nuclei which were used for all downstream analyses. Unsupervised clustering identified 24 cell clusters, which were then annotated with cell type via mapping against a reference dataset[9] and transferring cell labels (Fig. 1, Supplementary Dataset 1).

We then analyzed the data in a supervised, targeted approach, applying DAseq[21], a multiscale approach for detecting cell subpopulations with significant differential abundance (DA) between groups of interest (AD vs. Aged Controls). A DA population is defined as a cell subpopulation present at statistically higher or lower frequencies in one condition compared with another (AD vs Aged Control for this study). This approach supports rich, complex analyses within the context of observed cell subpopulations at the cohort level, in a manner that is not biased by inter-individual differences in cell type fractions. We identified a total of 9345 cells from 11 distinct DA subpopulations across all major cell type classes represented, ranging from 131 to 4505 cells in each population (Supplementary Dataset 1). The DA subpopulations were then collapsed into their respective cell type clusters and annotated as the DA cluster for each cell type

**Table 1 | Clinical, neuropathological, and demographic information for the 101 subjects from the Banner cohort profiled by snRNA-seq**

| Banner | | AD | Control |
|---|---|---|---|
| Total subjects | | 66 | 35 |
| Expired age | Mean | 85.9 | 83.5 |
| | SD | 8.7 | 8.2 |
| Sex | F | 33 | 17 |
| | M | 33 | 18 |
| Race | White | 65 | 35 |
| | Black | 1 | |
| PMI | Mean | 3.5 | 3.3 |
| | SD | 1.8 | 1 |
| Braak staging | I | | 6 |
| | II | | 11 |
| | III | 7 | 13 |
| | IV | 24 | 5 |
| | V | 16 | |
| | VI | 19 | |
| Plaque density | Zero | | 22 |
| | Sparse | 3 | 10 |
| | Moderate | 10 | 2 |
| | Frequent | 53 | 1 |
| CERAD-NP | Criteria not met | 1 | 21 |
| | Not AD | 3 | 11 |
| | Possible | 25 | 3 |
| | Definite | 37 | |
| NIA-R | Not AD | | 1 |
| | Low | | 1 |
| | Intermediate | 3 | |
| | High | 35 | |
| | Criteria not met | 28 | 33 |
| NIA-AA | Not AD | | 21 |
| | Low | | 14 |
| | Intermediate | 33 | |
| | High | 33 | |
| Clinical diagnosis | No CI | 20 | 28 |
| | MCI | 11 | 7 |
| | Dementia | 35 | |
| APOE | 22 | | 1 |
| | 23 | 9 | 10 |
| | 33 | 30 | 20 |
| | 24 | 3 | |
| | 34 | 22 | 4 |
| | 44 | 2 | |

*CERAD* semiquantitative measure of neuritic plaques, *NIA-R* NIA-Reagan diagnosis of AD, *NIA-AA* NIA-AA diagnostic guidelines for AD, *MCI* mild cognitive impairment.

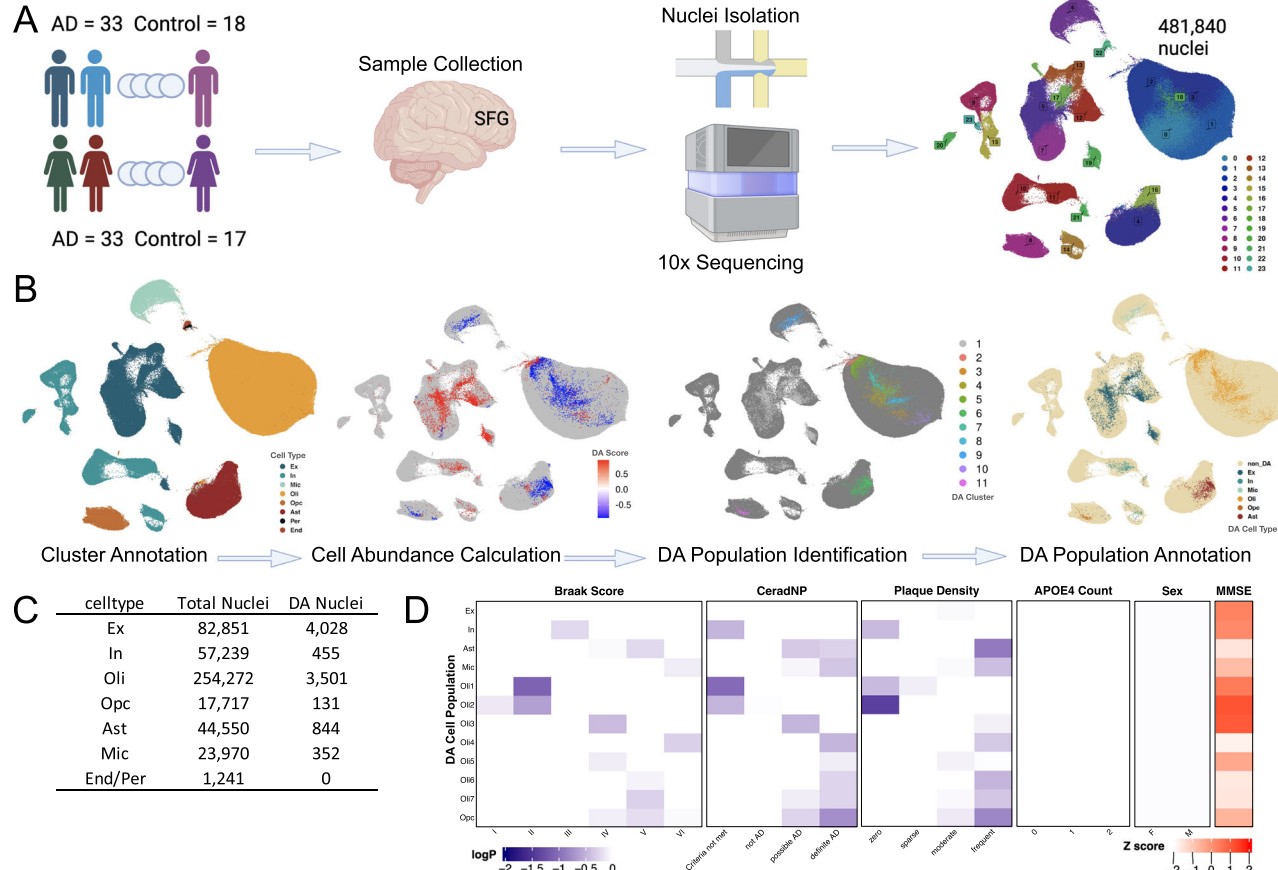

**Fig. 1 | Single nucleus RNA-seq from superior frontal gyrus samples from 101 human decedents. A** Experimental workflow for generation of transcriptomes from 481,840 nuclei. **B, C** Major cell type identification followed by detection of cell populations that are differentially abundant between AD and Control subjects. **D** With varying associations with clinicopathological AD traits. DA differentially abundant, SFG superior frontal gyrus, snRNA-seq single nucleus RNA-sequencing, Ex excitatory neurons, In inhibitory neurons, Mic microglia, Oli oligodendrocytes, Opc oligodendrocyte progenitor cells, Ast astrocytes. Panel **A** created with-BioRender.com released under a Creative Commons Attribution-NonCommercial-NoDerivs 4.0 Internationallicense.

(Fig. 1B, C, Supplementary Dataset 1). Top gene expression markers that discriminate DA subpopulations from non-DA cells of the same class are shown (Supplementary Dataset 2). We then examined whether individual DA cell subpopulations are associated with clinical and neuropathological traits of interest to AD (Fig. 1D). We observed that DA subpopulations over-represented among AD subjects (compared with Aged Controls) are primarily composed of glial cells and those over-represented among Aged Control subjects (compared with AD) are primarily composed of neuronal subtypes (Supplementary Dataset 1), consistent with the marked gliosis[22] and neuronal loss[23] that characterizes AD. Furthermore, the differential association of DA subpopulations with varying stages of neuropathological severity is consistent with the considerable changes in cell type fraction that are observed at different stages of AD.

We also observed that the identification of DA subpopulations offered advantages in detecting AD-linked genes beyond what we could identify through a more conventional differential expression analysis based on a cell-type-specific comparison of AD vs. Aged Control. For each major brain cell type, we generated differentially expressed genes (Supplementary Fig. 1, DEG, $|\log_2 FC| > 0.25$ and Bonferroni adjusted $p < 0.05$) identified via a comparison of DA cells against non-DA cells (DA, Supplementary Fig. 1, x-axes) as well as a comparison of AD cells against Aged Control cells (DX, Supplementary Fig. 1, y-axes). For all cell types, the range of DEG log2FC was larger for the DA-based comparisons, with a commensurately larger set of DEG identified exclusively using that comparison (Supplementary Dataset 2). This included DEG that was detected solely within a comparison

of DA against non-DA cells ('DA only') as well as comparatively fewer DEG detected only within a comparison of AD against Aged Control for each major cell type ('DX only'). We also observed a significant enrichment of AD GWAS risk loci prioritized genes[24] among the DA-only genes in microglia, inhibitory neurons, and OPC. A complementary approach using MAGMA[25] also identified an enrichment of AD GWAS risk genes among DA-only genes in microglia compared (Supplementary Dataset 2). Overall, these findings are supportive of the potential for DA based cell clustering to demarcate biologically informative cell groupings that might be used to frame further analyses.

## Cell type resolution of expression quantitative trait loci reveals a linkage between regulatory variants and neurological traits

We integrated genotype data of 99 subjects based on WGS from these 101 subjects with individual cell type expression profiles to determine cell type-specific expression quantitative trait loci (eQTL, Fig. 2A), which can then be directly investigated as well as used as inputs into downstream analyses including causal network modeling. We adopted a conservative approach to the detection of eQTL using snRNA-seq data which includes scran normalization, mean aggregation of expression across specific cells from one subject; incorporation of principal components as covariates in the associated linear mixed modeling, and accounting for multiple testing by using the conditional false discovery rate, leveraging the recently published eQTL summary statistics from the AMP-AD meta-analysis of bulk RNA-seq from cortical tissue as an external reference set[26]. We classified cis eQTL

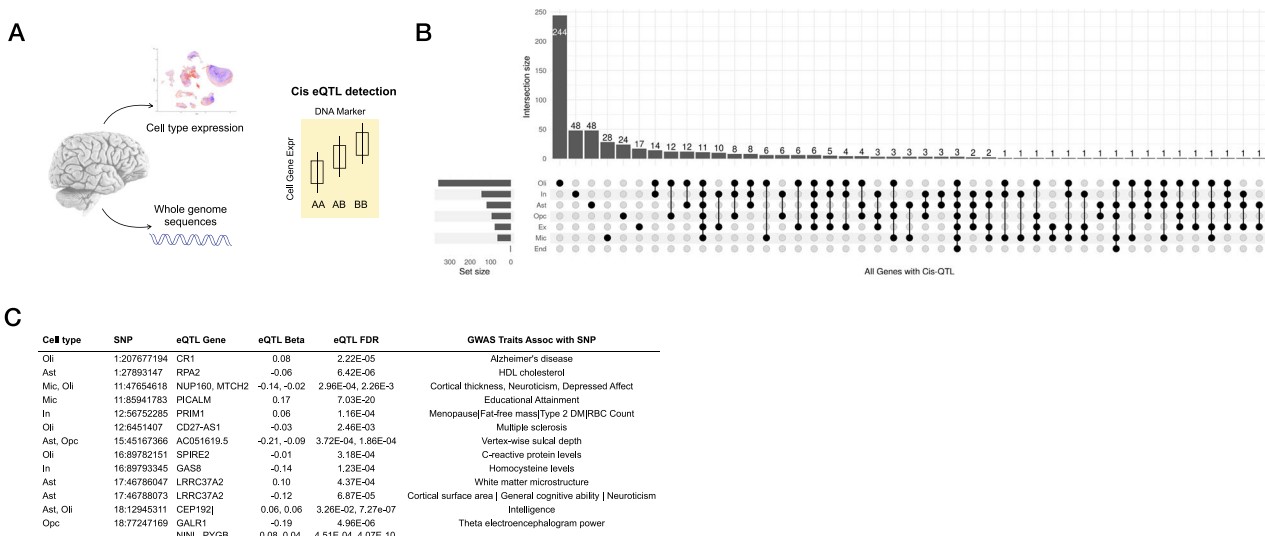

| Cell type | SNP | eQTL Gene | eQTL Beta | eQTL FDR | GWAS Traits Assoc with SNP |
|---|---|---|---|---|---|
| Oli | 1:207677194 | CR1 | 0.08 | 2.22E-05 | Alzheimer's disease |
| Ast | 1:27893147 | RPA2 | -0.06 | 6.42E-06 | HDL cholesterol |
| Mic, Oli | 11:47654618 | NUP160, MTCH2 | -0.14, -0.02 | 2.96E-04, 2.26E-3 | Cortical thickness, Neuroticism, Depressed Affect |
| Mic | 11:85941783 | PICALM | 0.17 | 7.03E-20 | Educational Attainment |
| In | 12:56752285 | PRIM1 | 0.06 | 1.16E-04 | Menopause\|Fat-free mass\|Type 2 DM\|RBC Count |
| Oli | 12:6451407 | CD27-AS1 | -0.03 | 2.46E-03 | Multiple sclerosis |
| Ast, Opc | 15:45167366 | AC051619.5 | -0.21, -0.09 | 3.72E-04, 1.86E-04 | Vertex-wise sulcal depth |
| Oli | 16:89782151 | SPIRE2 | -0.01 | 3.18E-04 | C-reactive protein levels |
| In | 16:89793345 | GAS8 | -0.14 | 1.23E-04 | Homocysteine levels |
| Ast | 17:46786047 | LRRC37A2 | 0.10 | 4.37E-04 | White matter microstructure |
| Ast | 17:46788073 | LRRC37A2 | -0.12 | 6.87E-05 | Cortical surface area \| General cognitive ability \| Neuroticism |
| Ast, Oli | 18:12945311 | CEP192\| | 0.06, 0.06 | 3.26E-02, 7.27E-07 | Intelligence |
| Opc | 8:77247169 | GALR1 | -0.19 | 4.96E-06 | Theta electroencephalogram power |
| Ast, Oli, Oli, Oli | 20:25349123 | NINL, PYGB, NINL, ABHD12 | 0.08, 0.04, 0.03, -0.03 | 4.51E-04, 4.07E-10, 3.84E-06, 1.38E-02 | Phosphatidylinositol (18:0_20:4) levels |
| Mic | 6:32644620 | HLA-DQB1 | 0.11 | 3.24E-05 | Epstein Barr virus nuclear antigen 1 IgG levels |

**Fig. 2 | Cell-type expression QTL detection in superior frontal gyrus.**
**A** Integration of whole genome sequences with cell type expression to detect cell type eQTL associations **B** with shared and cell type-specific distribution **C** revealing multiple cell type eQTL linking with DNA loci that have been implicated in diverse neurological and neurodegenerative traits via GWAS data[30]. eQTL expression quantitative trait loci, Ex excitatory neurons, In inhibitory neurons, Mic microglia, Oli oligodendrocytes, Opc oligodendrocyte progenitor cells, Ast astrocytes.

associations with an FDR ≤ 0.05 as significant, detecting a total of 1185 associations from 560 genes (Fig. 2B). Consistent with what has been observed[27,28], the majority of the eQTL associated genes (eGenes) were detected only in a single cell type ($n = 409$), supporting the utility of snRNA-seq in resolving regulatory relationships that may not be evident at the level of bulk tissue transcriptomics (Supplementary Dataset 3).

Expression QTL relationships have been determined to be enriched for disease risk genetics[29] and offer a means to contextualize and identify mediating mechanisms that link genetic variation with disease phenotypes. We thus annotated each eQTL locus with any published associations published by the NHGRI-EBI GWAS Catalog[30]. We observed many instances of eQTL loci which have been previously published in association with traits that link with AD and associated risk factors and comorbidities, such as cortical thickness, educational attainment, Type 2 diabetes mellitus, homocysteine levels, and HDL cholesterol levels (Fig. 2C).

## Alzheimer's GWAS locus rs679515 impacts *CR1* networks in oligodendrocytes, known erythrocyte regulators, and peripheral blood hematocrit

We observed that two reported AD risk loci, rs11118328[31] (chr1:207677194) and rs9429780[24,32] (chr1: 1:207493845) are both eQTLs for Complement Receptor 1 (*CR1*) expression in oligodendrocytes (Supplementary Dataset 4). Recently, Fujita et al.[27] and Mathys et al.[33] published analyses based on snRNA-seq from 433 and 427 dorsolateral prefrontal cortex samples respectively, from subjects included within the Religious Orders Study and Rush Memory and Aging Project (ROSMAP) cohort[34]. Using these data (Fujita-ROSMAP and Mathys-ROSMAP DLPFC snRNA-seq) we confirmed the association between both loci and *CR1* expression in oligodendrocytes (Supplementary Dataset 4). Fujita et al.[27] also reported the nearby most significant AD risk locus rs679515 (chr1:207577223, Major allele: C, Minor allele: T, Reference allele: T) as an eQTL in oligodendrocytes, although this variant had not emerged in our analysis. Examination of our eQTL results revealed that rs679515 had been removed following linkage disequilibrium-based pruning with rs9429780. We then confirmed that rs679515 was also an eQTL for *CR1* in oligodendrocytes in our dataset by a direct association test, in agreement with Fujita et al.[27] (Fig. 3A,

Supplementary Dataset 4), where minor allele T (also the AD risk allele) is associated with higher *CR1* expression. Out of the three loci under consideration (rs11118328, rs9429780, and rs679515), rs679515 was most strongly associated with CR1 expression within oligodendrocytes. Consistent with Fujita et al.[27], we also observed colocalization of the genetic signal driving *CR1* expression in oligodendrocytes with AD GWAS risk signal[35] within the Banner cohort (posterior probability: 0.985), with rs679515 emerging as the most likely causal variant explaining this shared signal (posterior probability: 0.817, Fig. 3B, Supplementary Dataset 4). Further examination of the association between these three loci and oligodendrocyte expression of *CR1* within the Banner SFG data revealed that this was driven by the association within AD subjects (Supplementary Dataset 4), indicating an AD-specific interaction for all three loci, but most strongly for rs679515 (AD diagnosis/dosage interaction $p$-value = 2.1e−5). We did not observe a significant AD-specific interaction within the Fujita-ROSMAP or Mathys-ROSMAP data, (Supplementary Fig. 2A, Supplementary Dataset 5). Whether the difference in detection of an AD-specific interaction between rs679515 and *CR1* expression is explained by differences in cohort composition or technical variation remains to be investigated.

Given the potential for cell type-specific eQTL associations to illuminate the biological context of risk-associated variants, we hypothesized that altered *CR1* expression mediates some fraction of the risk effect of this allele. We, therefore, constructed a targeted gene regulatory network aimed at identifying genes that are downstream of *CR1*, conditioned on the relationship with rs679515, using a causal inference testing approach[36] applied to the 66 AD samples. We identified 25 significantly associated downstream genes (FDR < 0.05, Fig. 3C, Supplementary Dataset 4) including Complement C3b/C4b Receptor 1 Like (*CR1L*) and Erythropoietin Receptor (*EPOR*). Molecular functional enrichments of the rs679515/*CR1* network revealed themes of erythrocyte biology and hematological traits (Fig. 3D, Supplementary Dataset 4), including enrichments for genes linked with erythrocyte signaling and morphology. Given the inclusion of *EPOR* and enrichments implicating hematological traits, we hypothesized that this may also manifest as differences in blood laboratory parameters within our study population. We performed a chart review of antemortem hematocrit values from 49 AD subjects within our study, collected during a time of comparatively stable health. Linear

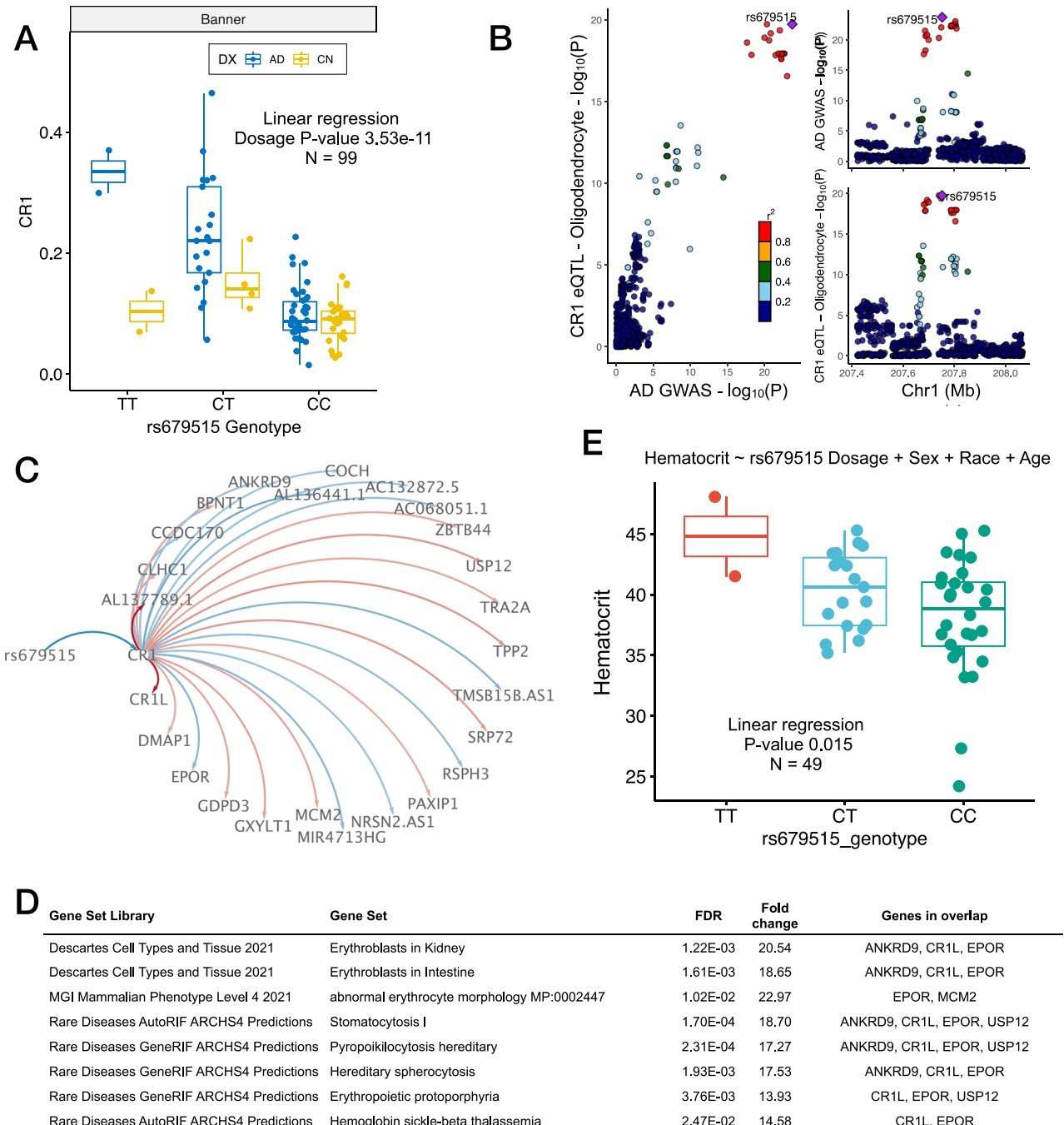

**Fig. 3 | GWAS loci rs679515 impacts CR1 in oligodendrocytes, known erythrocyte regulators and peripheral blood hematocrit. A** AD-specific eQTL association between AD GWAS loci rs679515 genotype and *CR1* expression within oligodendrocytes. **B** Colocalization of genetic signal underpinning AD GWAS risk and *CR1* expression within oligodendrocytes driven by rs679515. **C** Causal inference network illustrating conditional relationships between AD GWAS locus rs679515, *CR1* expression, and downstream molecular networks, including impacts on EPO receptor expression. **D** Molecular functional enrichments of the rs679515/*CR1* network revealed themes of erythrocyte biology and hematological traits. **E** Antemortem hematocrit levels from 49 Banner AD subjects, stratified according to rs679515 dosage. Enrichments based on Fisher's Exact Test. AD Alzheimer's disease, GWAS Genome-Wide Association Study, HCT hematocrit, FDR false discovery rate.

regression revealed a significant association (*p*-value: 0.015) between rs679515 major allele dosage and hematocrit (Fig. 3E, Supplementary Dataset 4) while adjusting for sex, race, and age at collection, where the major allele is associated with lower hematocrit in AD patients.

We then sought to determine whether rs679515 is also associated with antemortem hematological parameters (including hematocrit) within the ROSMAP cohort. We took the last available measurement within three years before death for each individual and examined the association of measurements with the rs679515 major allele dosage.

We observed significant associations with several hematological traits, including hematocrit, hemoglobin concentration (Hb), red blood cell count (RBC), and anemia diagnosis (Supplementary Dataset 5). These hematological traits were also significantly associated with a variety of neuropathologies, particularly neurofibrillary tangle burden. Interestingly, we observed a neuropathology-associated inversion of the association between rs679515 major allele dosage and these hematological traits (Supplementary Fig. 2B, Supplementary Dataset 5), whereby major allele dosage was positively associated with

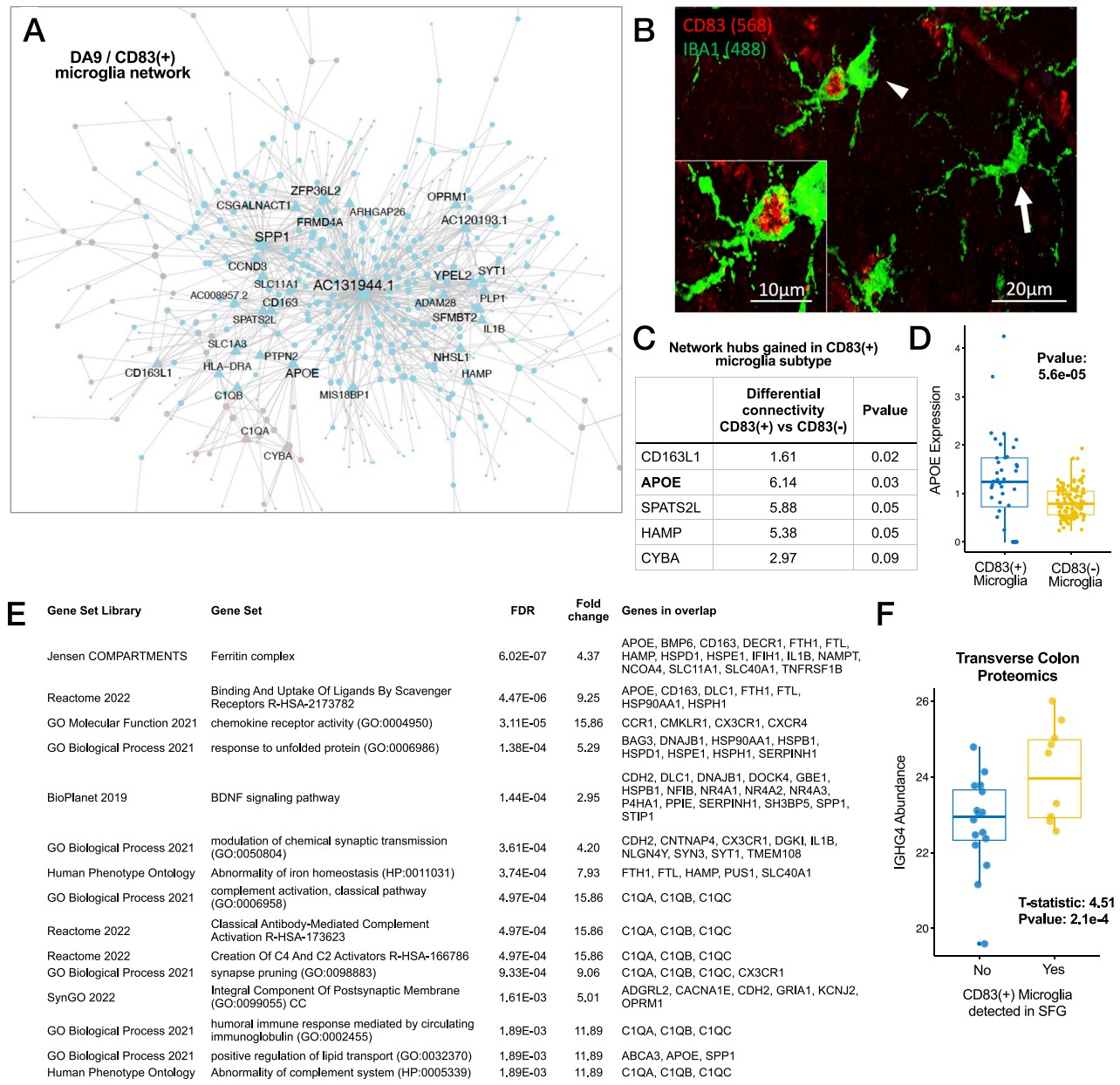

**Fig. 4 | Exploration of CD83(+) microglia. A** Multiscale gene coexpression network constructed from differentially abundant CD83(+) microglial subpopulation reveals novel antisense transcripts functioning as network hubs.
**B** Photomicrograph of CD83(+) microglia within Layer II of superior frontal gyrus in a Braak IV AD subject. CD83 reactive microglia appear (see insert) compared to unreactive microglia (arrow). **C** APOE gained as a network hub and **D** at increased

expression in CD83(+) microglia compared with CD83(−) microglia. The mean APOE expression for each subject (stratified by microglial type) is shown.
**E** Representative molecular functional enrichments among CD83(+) microglia hub genes. **F** Increased abundance of Immunoglobulin Heavy Chain protein IGHG4 protein in the transverse colon as a function of the presence of CD83(+) microglia in superior frontal gyrus, while controlling for AD, Age, Sex, and PMI (Samples $N = 26$).

hematocrit/RBC/Hb (and inversely associated with the presence of anemia) in subjects with low NFT burden (Braak stage: 0–II), and inversely associated with hematocrit/RBC/Hb (and positively associated with the presence of anemia) in subjects with high NFT burden (Braak stage: V–VI). Taken together, these findings suggest that rs679515 influences molecular networks across multiple tissues, with impacts on hematological traits that occur in an AD/neuropathology-specific manner.

**Multiscale network analysis of AD-associated cell types reveals molecular drivers in CD83(+) microglia**
We constructed cell type-specific gene regulatory networks using two complementary network modeling approaches including a

probabilistic causal network methodology (RIMBANet) and Multiscale Embedded Gene Expression Network Analysis (MEGENA) reviewed here[37]. We focused our initial efforts on the investigation of networks constructed from DA microglia subpopulation DA9 (Fig. 4), which revealed directed networks containing many known microglial-relevant transcripts including secreted glycoprotein and Type 1 immunity regulator *SPP1*, *TREM2*, and complement component 1q genes (*C1QA*, *C1QB*, *C1QC*). To better characterize the subpopulation, we compared the gene signatures present in DA9 to microglia signatures reported in the scientific literature[38]. Due to the large number of nuclei sampled in our study, many genes that are significantly differentially expressed in DA9 when compared with other nuclei in microglia demonstrate small log-fold changes (Supplementary

Dataset 2). We observed the downregulation of many homeostatic microglial genes, such as *TMEM119, P2RY12, P2RY13, CX3CR1,* and *SELPLG.* Conversely, we observed upregulation of many of DAM signatures genes, such as *SPP1, APOE, CD9, CLEC7A, FTH1,* and *FLT1* (Supplementary Fig. 3). Although *TREM2* and *TYROBP* are expressed in the DA9 microglia, they are not differentially expressed when compared with other microglia cells or stratified according to AD/Aged Control status. We also noted the overlap between DA9 differential expression and several dystrophic[39] and other microglia signatures[8,9] reported in the scientific literature (Supplementary Fig. 3). We observed cell surface marker *CD83* as the most strongly overexpressed transcript (Supplementary Dataset 6) among DA9 microglia. Herein we refer to DA9 microglia as CD83(+) microglia for brevity, however we note that these cells are characterized by several marker genes beyond *CD83* and that our data includes other microglia that express *CD83,* but which are not members of the same DA subpopulation. Despite this, given the reports of CD83(+) microglia within the scientific literature already[7,40], we reasoned that CD83(+) would be an informative, though imperfect label. We detected CD83(+) microglia in 47% of AD (*n* = 31 of 66) and 25% of Aged Control (*n* = 9 of 35) subjects profiled within the Banner SFG snRNA-seq study.

Key driver analyses revealed the most influential network node to be an antisense transcript *AC131944.1* (ENSG00000286618.1, Fig. 4A), which, to our knowledge, has not been described in the context of microglial biology or AD, but which is encoded antisense to *SPP1.* In addition, we observed *APOE* as a key network driver in CD83(+) microglia, whereas we did not observe it as a network driver in non-CD83(+) microglia using a permutation-based network comparison approach (Fig. 4C). In addition to its gain of network influence in CD83(+) microglia, we also observed greater expression of *APOE* in these cells compared with CD83(−) microglia (Fig. 4D).

To understand whether CD83(+) microglia are transcriptionally adapted to specific biological pathways, we performed a gene set enrichment analysis upon the set of 308 hub genes (genes with a MEGENA connectivity score of ≥4, Fig. 4E, Supplementary Dataset 7) computed against a background of 3083 genes that were robustly expressed among the full set of detected microglia (>5% of the cells) within the study. We observed enrichments suggestive of complement activation, ferritin, and iron processing, antibody immune response, and lipid processing.

We then compared CD83(+) microglia transcriptomic signatures against a recent snRNAseq study of human microglia in AD[41] (Supplementary Dataset 7). Prater et al.[41] applied a trajectory mapping approach to microglia-enriched snRNA-seq generated from 22 postmortem DLPFC samples and identified 10 microglial clusters with distinct transcriptomic, molecular pathway, and phenotypic profiles. We observed striking enrichments against two microglial clusters in particular: Cluster 5 (which contains CD83 and is primarily enriched for stress-autophagy and endolysosomal network pathway activity, *p*-value: 3.51E−267, Odds Ratio: 158.8) and Cluster 9 (primarily enriched for senescence, iron homeostasis and cytokine production pathway activity, *p*-value: 3.46e−161, Odds Ratio: 50.1). In combination with the direct enrichment of CD83(+) microglia drivers indicating perturbed iron processing (Fig. 4E), this may suggest that our reported CD83(+) microglia are actually a senescent type of microglia[42].

## CD83(+) microglia are also detected in two independent single-nucleus RNA-seq data sets collected from the dorsolateral prefrontal cortex

To evaluate whether CD83(+) microglia are also present in additional snRNA-seq data sets, we again accessed the Fujita-ROSMAP[27] and Mathys-ROSMAP[33] DLPFC snRNA-seq and performed DAseq using the same workflow we had applied to the Banner SFG snRNA-seq. In both data sets, we identified an AD-associated CD83(+) microglial subpopulation that expressed marker genes that extensively overlap the

Banner SFG snRNA-seq-defined CD83(+) microglial marker genes. In a comparison of CD83(+) microglia and non-CD83(+) microglia, we identified 597 within the Banner SFG (| log₂FC | > 0.5, Bonferroni corrected *p* < 0.05), 312 DEG within the Fujita-ROSMAP DLPFC and 278 DEG within the Mathys-ROSMAP snRNA-seq. Banner CD83(+) microglia DEG overlapped significantly with both Fujita-ROSMAP CD83(+) microglia DEG (*p*-value: 4.04e−120, Odds ratio: 28.2, Fisher's Exact Test, Supplementary Dataset 8, Supplementary Fig. 4) and Mathys-ROSMAP CD83(+) microglia DEG (*p*-value: 1.69e−74, Odds ratio: 18.3, Fisher's Exact Test, Supplementary Dataset 8, Supplementary Fig. 4). This included shared overexpression of many genes that had emerged during the Banner SFG network analysis, including *CD83, AC131944.1, SPP1, APOE* and *ATP1B3.* Within the Fujita-ROSMAP (Mathys-ROSMAP) data we observed CD83(+) microglia in 25% (38%) of AD subjects and 4% (24%) of Control subjects.

## CD83(+) microglia are associated with increased multi-regional neuritic plaque and neurofibrillary tangle burden

We next sought to understand whether there may be any technical, demographic, clinical, or neuropathological differences between these groups. Within the Banner SFG snRNA-seq, we did not observe any difference in age of death, sex, ethnicity, post-mortem interval (PMI), neuritic plaque density, neurofibrillary tangle burden, clinical dementia rating or ApoE4 carriage rates between AD subjects with and without CD83(+) microglia (Supplementary Dataset 9). We also inspected whole-body autopsy results available for 68 (AD *n* = 45, Aged Control *n* = 23) of the 101 Banner SFG subjects, which allowed us to examine potential associations with diverse disease comorbidities. We did not identify any difference in rates of Type 2 diabetes mellitus, hypertension, obesity, atrial fibrillation, coronary stenosis, cardiomegaly, atherosclerosis, smoking, or chronic obstructive pulmonary disease (Supplementary Dataset 9).

Given that CD83 is a marker of mature dendritic cells, with complex, bidirectional interactions between CD83 and diverse pathogens[43–45], we hypothesized that CD83(+) AD subjects may differ from CD83(−) AD subjects on the basis of microbial or immunological perturbation which could be occurring peripherally, including during end-stage dementia. Aspiration pneumonia and subsequent sepsis are common causes of death in subjects that die from AD[46], with autopsy-confirmed pneumonia present in 54% of subjects that died with AD and 49.6% of all subjects within the BBDP[47]. We did not observe associations between autopsy-confirmed pneumonia and the presence of CD83(+) microglia among AD subjects (*p*-value: 0.37, Odds Ratio: 0.57, Fisher's Exact Test).

We then examined potential clinicopathological associations within the Fujita-ROSMAP and Mathys-ROSMAP snRNA-seq samples. Together, these data sets comprised 366 unique subjects with AD, of which 28.4% had CD83(+) microglia detected (detected: *n* = 104, not detected: *n* = 262). We examined associations for each data set separately, as well as combined (Supplementary Dataset 9, Supplementary Fig. 5) observing similar trends of significantly increased global AD pathology burden (a quantitative summary across five brain regions of neuritic plaque, diffuse plaque, and neurofibrillary tangle density) among AD subjects with CD83(+) microglia compared with CD83(−) AD subjects (*T*-statistic: 4.23 *p*-value < 2.9e−5), which was driven by an increase in neuritic plaque density (*T*-statistic: 4.26 *p*-value: 3.4e−5) and neurofibrillary tangle density (*T*-statistic: 4.16 *p*-value: 4.9e−5), but not diffuse plaque density (*T*-statistic: 0.37 *p*-value: 0.79). These significant associations were also observed in a complementary linear modeling approach, looking across samples from all 618 unique subjects while adjusting for age of death, sex, and AD diagnosis (Supplementary Dataset 9). Subsampling of the ROSMAP DLPFC snRNA-seq data to sets of 66 AD subjects (the size of the Banner SFG data, where we did not detect clinicopathological associations) resulted in our detection of increased (FDR < 0.05) global AD pathology burden in

only 12% of subsamplings and detection of increased neuritic plaques in 10% of subsamplings (Random subsamplings, $n = 1000$). These findings are consistent with an improved power to detect moderate clinicopathological association with CD83(+) microglia in AD subjects within the larger ROSMAP DLPFC snRNA-seq data, rather than necessarily implying a discordance across brain regions or cohorts.

### The transcriptional regulatory model of *CD83* in microglia implicates transcription factor *NR4A2*

To better understand the potential transcriptional context that may mediate the expression of CD83 in microglia, we constructed a transcriptional regulatory model for *CD83* as learned from independent microglial transcriptomic data. We extracted expression levels of *CD83* and all candidate transcription factors from a previously published bulk, sorted microglial RNA-seq dataset[48]. These data were then integrated with putative transcription factor binding sites for *CD83* and Transcriptional Regulatory Network Analysis[49] (Trena) was used to identify which transcription factors best predict CD83 expression. The strongest candidate regulator of *CD83* expression to emerge was steroid-thyroid hormone-retinoid superfamily member Nuclear Receptor Subfamily 4 Group A Member 2 (*NR4A2*). Subsequent investigation of *NR4A2* expression in CD83(+) microglia within our data revealed significant upregulation compared with CD83(−) microglia ($\log_2FC = 1.17$, FDR = 5.42e−61) and examination of our CD83(+) microglial networks revealed direct connectivity between *CD83* and *NR4A2* (Supplementary Fig. 6A).

To further substantiate our hypothesis that *NR4A2* plays a pivotal role as a transcriptional regulator of *CD83* in microglia, we conducted a systematic assessment of transcription factor (TF) activity within regulatory regions of *CD83*. This analysis was performed using previously published microglial ATAC-seq data[48] and included comparisons with oligodendrocytes, GABAergic neurons, and glutamatergic neurons[50]. Initially, we employed an activity-by-contact approach[51] to predict regulatory regions, followed by the TOBIAS footprinting method[52] to estimate TF binding within those regions. Among the five regulatory regions associated with *CD83*, we identified the presence of bound *NR4A2* motif in the microglia-specific open chromatin region located 22 kb upstream of *CD83* (Supplementary Fig. 6B). We further confirmed the regulatory relationship between this open chromatin region and CD83 by calculating a strong correlation (Spearman $\rho = 0.44$) between their chromatin accessibilities (Supplementary Fig. 6C, D). Together, these observations support the hypothesis that *NR4A2* is a significant transcriptional regulator of *CD83* within microglia.

### CD83(+) microglia are associated with increased immunoglobulin IgG4 in the transverse colon

We then hypothesized that a potential microbial perturbation could be reflected in the proteome of the transverse colon (TC), the anatomical site with the highest microbial diversity[53]. We generated mass spectrometry proteomics data from frozen TC samples from a subset of 26 subjects with SFG snRNA-seq. We examined whether any proteins are differentially abundant as a function of the presence of CD83(+) microglia within the SFG while adjusting for AD status, age of death, sex, and PMI. The most differentially abundant protein in the transverse colon of subjects with CD83(+) microglia was IGHG4 (Immunoglobulin Heavy Constant Gamma 4, Fig. 4F, *T*-statistic: 4.5 *p*-value: 2.1e−4, Supplementary Dataset 10) which forms the constant region of the IgG4 antibody heavy chain. This observation was suggestive of increased IgG4 tissue response in the TC wall of subjects with CD83(+) microglia and more broadly, consistent with a potential microbial interaction (whether direct or indirect) between components of the gut microbiome and the presence of CD83(+) microglia. In addition to IGHG4, we also observed an increased abundance of immunoglobulin heavy chain components IGHG2 and IGHG3, which comprise IgG2 and IgG3 immunoglobulins respectively (Supplementary Dataset 10).

## Discussion

In this study, we introduce a publicly available, postmortem multiomic data set generated from 101 subjects enrolled in the BBDP[19]. Through the integration of snRNA-seq from SFG, coupled with WGS data and extensive neuropathological and antemortem characterizations, we report a powerful resource for the exploration of cell specific transcriptomics and gene networks in the context of aging and AD.

We adopted an approach for identifying cell subtypes that are differentially abundant between AD and Aged Control subjects. We observed that this method was associated with an improved power to detect DEGs versus a conventional comparison of AD against Aged Controls, with these genes demonstrating a statistical enrichment for AD GWAS associated genes within microglia, inhibitory neurons and OPC populations.

We found that integration of WGS data with snRNA-seq data yielded substantial eQTL associations, including in minor cell fractions such as microglia, and thus which are unlikely to have been detected in bulk transcriptomics. Many eQTL loci were also previously linked with diverse neurological and AD-associated traits in published genetic studies and thus represent opportunities to contextualize complex genetic-phenotypic associations through cell-specific gene regulatory relationships. In agreement with Fujita et al.[27], we observed that a CR1-associated AD GWAS locus also contains an eQTL signal for *CR1* in oligodendrocytes, that these genetic signals colocalize and are likely driven by rs679515. We observed complex impacts of rs679515 upon hematological molecular networks and peripheral blood hematocrit, RBC count, hemoglobin concentration, and anemia diagnosis. Interestingly, these associations appear to occur in a neuropathology-specific manner.

*CR1* has been associated with AD via several polymorphisms[54–57], though findings have been variable across datasets[58], and clarifying whether the pathogenic mechanism occurs in the central nervous system (CNS) or periphery remains elusive. *CR1* encodes a transmembrane glycoprotein receptor for multiple ligands (including complement components *C3b* and *C4b*) and is abundantly expressed on erythrocytes and leukocytes[59], though also on tissue-resident macrophages and dendritic cells and on multiple brain cell types, including microglia, astrocytes[60], and oligodendrocytes[61]. Peripherally, C3b-opsonized immune complexes (including amyloid aggregates) bind to *CR1* on erythrocytes and undergo eventual phagocytic clearance in the liver and spleen[59], representing a major hypothesized mechanism that links *CR1* genetic variants with AD risk[62,63]. Despite this notion, plasma *CR1* levels have not been reported to be altered in AD, though are modestly increased in carriers of several *CR1*-linked AD GWAS risk variants[64]. We observed the rs679515/*CR1* eQTL association only in oligodendrocytes, with network findings that were sufficiently reminiscent of canonical erythrocyte regulators to motivate our investigation of effects upon peripheral hematocrit. We also note recent reports that rs679515 is a protein QTL for plasma CR1 abundance[65]. Overall, our findings align with a potentially multi-tissue impact of rs679515 / *CR1*, although whether the oligodendrocyte rs679515/*CR1* network actually reflects a key AD pathomechanism or is just conserved enough to suggest the relevant peripheral mechanism remains unresolved. The effects of rs679515 on both hematological traits and oligodendrocytes are consistent with a robust genetic impact upon CR1 abundance across diverse tissues and need not necessarily imply a unified cross-tissue disease pathomechanism. However, the GWAS loci for *CR1* have also been independently found to be co-localized in oligodendrocytes in AD by an integrative analysis of multiple snRNA-seq datasets from several different brain regions[28]. We also note recent reports by Yang et al.[4] on the AD-relevant activity of *CR1* within perivascular macrophages (PVM). Given complex reciprocal interactions between oligodendrocytes and the neurovascular unit[66,67], simultaneous evaluation of *CR1* and AD GWAS loci in PVM and oligodendrocytes may illuminate a shared mechanism that could

explain the observed CNS as well as hematological effects in a parsimonious manner. Further study of whether peripheral CR1 levels are associated with rs679515 in an AD-specific manner may also be an informative direction for future investigation.

We observed an AD-associated, CD83(+) microglial subtype, which was present in 47% of neuropathologically defined AD subjects within the Banner SFG cohort, 25% of the Fujita-ROSMAP cohort, and 38% of the Mathys-ROSMAP cohort. Gene regulatory network reconstruction based on these microglia identified a gain in network connectivity of APOE and the presence of multiple microglial nodes of interest, including TREM2 and complement C1Q complex genes, with the most influential network drivers comprised of comparatively understudied noncoding RNA gene AC131944.1/ENSG00000286618.1, which is encoded antisense to critical microglial regulator SPP1[68,69], with a positive regulatory relationship between AC131944.1 and SPP1 implied by our CD83(+) microglial networks. Functional enrichments of CD83(+) microglial genes highlighted biological themes of humoral immunity, complement activity, lipid processing, and iron transport. Further comparison against AD microglial snRNA-seq signatures[41] was suggestive of autophagy, endolysosomal networks, and microglial senescence. A transcriptional model constructed from independent microglial transcriptomic data revealed NR4A2 as the most likely transcription factor contributing to CD83 expression within the CD83(+) microglia. NR4A2 (also known as NURR1) has been well described in dopaminergic neurons, particularly in the context of Parkinson's Disease[70], though is also responsive to LPS activation in cultured microglia[71] and linked with cognitive and neuropathological changes in the 5XFAD mouse model of AD[72,73]. Previous studies have reported CD83(+) microglia as AD-associated[7] and recent work also links CD83 as a marker of microglia engaged in myelin debris phagocytosis as well as a potential modulator of autoimmune neuroinflammation during certain disease states[40]. These observations are also consistent with reports of microglial and astrocytic NR4A2 activity restraining neuroinflammation-induced cell death[74].

Within the Banner-SFG AD subjects, the presence of CD83(+) microglia was not differentially associated with PMI, demographic factors, major disease comorbidities, or terminal pneumonia infection. The presence of CD83(+) microglia was associated with a significant increase in multi-regional neuritic plaque and neurofibrillary tangle density in both Fujita-ROSMAP and Mathys-ROSMAP, though not within Banner-SFG. We also observed a significantly increased abundance of IGHG4 protein within the transverse colon of Banner-SFG subjects which is suggestive of increased IgG4 immunoglobulin. IgG4 is one of four main IgG immunoglobulin subclasses and the least prevalent among healthy adults[75] (approximately 5% of IgG) with important structural differences driving functional differences, including reduced triggering of antibody-dependent cell-mediated phagocytosis and complement activation[76]. Although IgG4 is generally regarded as exerting a relatively anti-inflammatory effect relative to other IgG subclasses, it is still capable of driving phagocytosis of opsonized microbial antigens[77] and mediating complement activation at higher antigen and antibody concentrations[78]. Despite the emergence of IGHG4 as the most differentially abundant protein, we did also observe a significant upregulation of IGHG3 and IGHG2 (which would be expected to induce more aggressive antipathogen responses), which is overall suggestive of a complex, adaptive immune response present in the transverse colon of subjects with CD83(+) microglia in the SFG. It is intriguing to note that gene set enrichments generated on CD83(+) microglial hub genes were also reflective of the biology underpinning humoral immunity.

Our study has several limitations, including some inherent to snRNA-seq, such as the absence of expression for transcripts that are not abundantly present in the nucleus, and thus represents a major limitation in the scope of our transcriptomic findings. We also note that our primary findings are based on SFG samples, which was

selected for relevance to AD pathobiology[79], however the extent to which our findings might apply to additional brain regions is not yet clear. Despite this, the shared observations of CR1 eQTL and hematological associations within oligodendrocytes and CD83(+) microglia within the ROSMAP DLPFC snRNA-seq data is encouraging that these findings may apply to additional regions. In addition, gene regulatory network modeling approaches applied to snRNA-seq data are a rapidly evolving frontier, and best standards and practices are yet to emerge[37].

Our study demonstrates the power of combining multiomic, multi-tissue, and systems biology approaches to illuminate disease biology. These efforts were enabled by several key aspects of the Banner Sun Health Research Institute's Brain & Body Donation Program, including high tissue quality, detailed antemortem (e.g., hematocrit laboratory values), and postmortem characterizations and the availability of tissue from additional extra-CNS anatomical sites (e.g., transverse colon). An additional advantage of our study is the comparatively large sample size of 101 subjects, which supported well-powered, cell-specific eQTL analyses and the application of subsequent network modeling approaches, coupled with the opportunity to examine independent, complementary snRNA-seq data[27,33] to replicate and extend key findings. In addition to the specific findings described within this study, we encourage access to the associated transcriptomic, genetic, phenotypic and multiscale network data by interested investigators and hope that these resources can prove useful for the scientific community.

## Methods

### Brain tissue sample processing

Subjects were all volunteers in the Arizona Study of Aging and Neurodegenerative Disorders (AZSAND), a longitudinal clinicopathological study of aging, cognition, and movement in the elderly since 1996 in Sun City, Arizona. Autopsies are performed by the Banner Sun Health Research Institute Brain and Body Donation Program[19] (BBDP; www.brainandbodydonationprogram.org). All subjects sign Institutional Review Board-approved informed consents allowing both clinical assessments during life and several options for brain and/or bodily organ donation after death. Most subjects are clinically characterized with annual standardized test batteries consisting of general neurological, cognitive and movement disorders components, including the Mini Mental State Examination (MMSE). Subjects for the current study (Table 1; n = 101) were chosen by searching the BBDP database for a full spectrum of AD neuropathology, in the absence of other neurodegenerative disease diagnoses.

The complete neuropathological examination was performed using standard AZSAND methods[19]. The neuropathological examination was performed in a standardized manner and consisted of gross and microscopic observations, the latter including assessment of frontal, parietal, temporal, and occipital lobes, all major diencephalic nuclei and major subdivisions of the brainstem, cerebellum, and spinal cord (the lattermost only for those with whole-body autopsy). Detailed clinical data, postmortem neuropathological data, and demographics of the cohort are described in Table 1.

### Chromium 10x snRNA-seq

Superior frontal gyrus tissue (~50 mg) was homogenized in 1 ml of Nuclei Lysis buffer [Nuclei EZ Lysis Buffer (Sigma-Aldrich, St. Louis, MO, USA) supplemented with 1× cOmplete™ Protease Inhibitor Cocktail (Sigma-Aldrich, St. Louis, MO, USA) and RNasin Plus (Promega)] 10–15 times using pestle A "loose" followed by pestle B "tight" 10–15 times (DWK Life Sciences, Millville, NJ, USA). Homogenate was passed through a 70 μm 1.5 ml mini strainer (PluriSelect, El Cajon, CA, USA) and centrifuged at 500 rcf for 5 min at 4 °C. Nuclei pellet was resuspended in an additional 1 ml of Nuclei Lysis buffer and incubated for 5 min followed by centrifugation at 500 rcf for 5 minutes at 4 °C. 500 μl of 1× wash/resuspension buffer (1× PBS, BSA 2%, 0.2 U/μl RNasin

Plus) is added to the nuclei pellet and incubated for 5 min to allow adequate buffer exchange followed by centrifugation at 500 rcf for 5 min at 4 °C, repeated once more and resuspended in 500 µl of 1× wash/resuspension buffer. Resuspended nuclei were incubated with 1–2 drops of NucBlue Live ReadyProbes Reagent (ThermoFisher Scientific) and immediately sorted using the DAPI channel on the Sony SH800S (Sony Biotechnology, San Jose, CA, USA) with a 100 µm chip.

Nuclei were sorted for 15,000 events directly into 10 × 3′ v3.1 RT Reagent Master Mix and immediately processed with the 10× Genomics Chromium Next GEM Single Cell 3′ v3.1 (Dual Index) kit (10× Genomics, Pleasanton, CA). To minimize batch effects, each 10× chip contained samples from all disease and control groups. Samples were loaded, cDNA amplified, and the library constructed following the manufacturer's protocol. Library quality control (QC) was based on Agilent Tapestation 4200 HS D1000 screentapes (Agilent Technologies, Waldbronn, Germany). Multiplexed library pool was based on HS D1000 and Kapa Library Quantification Kit for Illumina platforms (Kapa BioSystems, Boston, MA) and sequenced at shallow depths on Illumina's iSeq 100 v2 flow cell for 28 × 10 × 10 × 90 cycles for estimated reads per cell. After demultiplexing, libraries were rebalanced based on reads per cell. Normalized pool QC was based on Agilent Tapestation 4200 HS D1000 and Kapa Library Quantification Kit for Illumina platforms and high-depth sequenced on Illumina's NovaSeq 6000 S4 v1.5 flow cell for 28 × 10 × 10 × 90 cycles.

### snRNA-seq data preprocessing, filtering, integration, clustering, and cell type annotation

Gene counts were obtained by aligning reads to the hg38 reference genome (GRCh38.p5, GCA_000001405.20) using CellRanger (v6.0.2) (10x Genomics)[80], with all default parameters adding the "--include-introns" flag. All the subsequent preprocessing was accomplished in the standard Seurat (v4.0) workflow[81] on each subject individually, with a set of rigorous quality control measures. Briefly, nuclei were removed with gene counts less than 200, having >5% mitochondrial, >5% ribosomal, or >0.1% hemoglobin counts. Nuclei with either too high or low total UMI counts were also removed, by fitting a sigmoid function to the distribution of $\log_{10}$UMI and setting the cutoff to the ln4 * scale factor away from the inflection point. Nuclei with $\log_{10}$GenesPerUMI < 0.8 were also removed. Finally, doublets were identified by DoubletFinder (v.3.0)[82] and removed. Each Seurat object was then normalized using the SCTransform[83] protocol. We regressed out, during the normalization, the number of genes, the number of UMIs, the percentage of mitochondrial genes, and cell cycle scores. Principal components (PC) were calculated using the first 3,000 variable genes, and Uniform Manifold Approximation and Projection (UMAP)[84] analysis was performed with the top 30 PCs.

Reference-based integration was employed for the integration of the 101 subjects in Seurat, using one male and one female as references. Reciprocal PCA (RPCA) was applied to the first 30 PCs of the 3,000 variable genes to find the anchors before integration by SCTransform. PCA and UMAP analysis were performed on the integrated object. Clustering was done using a resolution of 0.5 by the default Louvain algorithm. We annotated the cell clusters by leveraging the snRNA-seq data published earlier[9] as a reference (syn18485175), by cell type label transfer using the "FindTransferAnchors" and "TransferData" functions in Seurat and projecting query cells onto reference UMAPs.

### Single-nucleus expression quantitative trait locus (sn-eQTL) mapping

We ran whole genome sequencing for 103 individuals from the BBDP, 99 of which overlap with those in snRNA-seq. Reads were aligned to hg38 (GCA_000001405.15) using the BWA-mem aligner (v0.7.10)[85]. Variants were called using the GATK Best Practices workflow (v4.0.1.2)[86]. Variants with a minor allele frequency <0.05, with missing call rates >0.1, or having Hardy–Weinberg equilibrium exact test p-value < 1e−6 were removed from further analysis. Common variants for all the remaining biallelic sites were imputed in Beagle (v5.0)[87] using 1000 Genomes Phase 3 reference genotypes[88].

We followed the recommended workflow as reported recently[89] to map the sn-eQTL in this study. Specifically, for each annotated cell type, normalization was first done in scran (v1.24.0)[90] for all the nuclei within the cluster(s), followed by mean aggregation of the expression profiles across nuclei from each individual. Association tests were performed using linear models in LIMIX (v2.0.3)[91], including age, gender, race, disease status, and top 20 principal components as covariates. Cis-eSNPs were reported for those sites located within 250 kb of the transcribed region for a gene. The p values were corrected for multiple testing by using the conditional false discovery rate (cFDR) method[92], leveraging the meta-analysis eQTLs from cerebral cortical bulk tissues (syn16984815)[26] to increase power. The meta-analysis eQTLs were lifted up to hg38 by CrossMap (v0.6.3)[93].

### Differentially abundant (DA) cell population identification

Cell subpopulations whose abundance differs between the two states (AD and control in our study) were detected by DAseq[21], using the integrated Seurat object. The top 40 PCs of the expression profiles were used for comparison. The values of k used for the calculation of the score vector with kNN were set to 100–4000 with an increment of 500. DA measure value thresholds were set at −0.8 to 0.8. The clustering parameter was set to 0.01. The identified DA clusters were integrated with cell cluster annotation for their respective annotations.

### Cluster trait associations

To make an unbiased comparison, we collapsed the nuclei from the same individual in each cluster into one and annotated their phenotypic and neuropathological traits by the metrics from the corresponding individuals. The overrepresentation of individuals in each cluster was assessed as reported[9]. For categorical traits, enrichment was evaluated using the hypergeometric distribution (Fisher's exact test) and FDR correction over all clusters. Enrichment or depletion of quantitative traits was assessed individually by contrasting the average observed value across the individuals of a given cluster with a corresponding null distribution estimated by a scheme of 10,000 permutations. The deviation of the observed value from the random expected distribution was quantified using a z-score.

### Differential gene expression analysis

Differential expression of genes between conditions was identified by the FinderMarkers function of the Seurat package, using the MAST algorithm[94]. The adjusted p-value is based on Bonferroni correction using all genes in the dataset. For each cell type, two comparisons were made. One is between all the cells from AD subjects and those from control subjects (DX). The other comparison is between all the cells from DA populations and those from non-DA populations (DA). Only genes that are detected in a minimum fraction of 1% of cells in either of the two conditions were tested. Lists of DEGs were generated by filtering all genes for absolute $\log_2$fold change >0.25, adjusted p < 0.05.

### MAGMA gene set analysis

GWAS summary statistics from Bellenguez et al.[24] were downloaded from the European Bioinformatics Institute GWAS Catalog (https://www.ebi.ac.uk/gwas/) under accession no. GCST90027158. Genes were annotated using NCBI gene annotation for GRCh38. Gene sets identified by the two different contrast approaches (DA and DX) were tested using MAGMA (v1.10)[25]. The annotation window sizes were set as 100, 20, and 150, 50, respectively.

## Colocalization analysis

All genetic variants within 250 kilobases of the CR1 gene body were used in the colocalization analysis. AD GWAS risk summary statistics were obtained from Wightman et al.[35]. CR1 cis-eQTL association statistics were generated to include all variants without LD-based pruning used in our main cis-eQTL analyses. A set of 1122 variants which were present in both AD GWAS and CR1-eQTL analyses, were used for the analysis. Colocalization was performed using the R package *Coloc*[95] using the coloc.abf function. Default prior probabilities for colocalization were adopted, and minor allele frequencies were estimated from the Banner-SFG cohort. The total sample size for the AD GWAS summary data was set at $n = 34,720$ and the case ratio as 0.087.

## MEGENA and RIMBANet gene regulatory network construction

We ran gene network analysis for all the DA cell populations in each annotated cell type, following the recommended workflow[37]. Specifically, we constructed gene co-expression and detected multiscale gene modules using MEGENA[96] and Bayesian networks (BN) using RIMBANet[97]. For comparison, we chose the same number of nuclei in both AD and control cell populations and constructed the comparative networks using the same workflow. The nuclei were chosen based on the shortest Euclidean distances of the top 40 PCs between the expression profiles of DA and target populations. Differential connectivity scores for the nodes in the networks were calculated based on the procedure in the R package dnapath[98] (https://github.com/tgrimes/dnapath) with the p values generated by a scheme of 1,000 permutations.

## Construction of rs679515/CR1 gene regulatory network

We performed causal inference testing[36], to build a causal gene regulatory network focused on *CR1* in post-mortem SFG oligodendrocytes. This approach requires paired gene expression and genotype data for a large number of samples to establish the direction of regulation between *CR1* and its correlated genes. Causal inference testing (CIT) has been well-described previously[36]. Briefly, it offers a hypothesis test for whether a molecule (in this case, the expression of *CR1*) is potentially mediating a causal association between a DNA locus (rs679515), and some other quantitative trait (such as the expression of genes correlated with *CR1* and rs679515). Causal relationships can be inferred from a chain of mathematic conditions, requiring that for a given trio of loci (L), a potential causal mediator, i.e., *CR1* (G) and a quantitative trait (T), the following conditions must be satisfied to establish that G is a causal mediator of the association between L and T:

(a)    L and G are associated
(b)    L and T are associated
(c)    L is associated with G, given T
(d)    L is independent of T, given G

We used the R software package "cit"[99], to perform the causal inference test, calculating a false discovery rate using 1000 test permutations. Trios with a *Q* value < 0.05 were classified as significant, and the associated T genes were considered downstream of *CR1*.

## Transcriptional regulatory model of *CD83*

A transcriptional regulatory model of *CD83* was created as follows. Expression levels of *CD83* and all candidate transcription factors came from a previously published bulk, sorted microglial RNA-seq dataset[48] (syn25671134) and bulk, sorted GABAergic, glutamatergic, and oligodendrocytes (GEO GSE143666). The putative regulator region of *CD83* was defined using regions identified by GeneHancer[100]. With the regions identified by GeneHancer, FIMO was used to identify all putative transcription factor binding sites. We then generated a list of candidate transcription factors based on the presence of these transcription factor binding sites with the GeneHancer regions. Trena[85], which uses several regression and machine learning approaches to identify which transcription factors can best predict the expression of *CD83*, was used to generate the model. To ascertain the binding status of transcription factors across various brain cell types and in AD case/control subjects, we conducted footprinting analysis with the TOBIAS tool[101]. Building upon the settings employed in our previous study[102], we examined 431 motifs, representing 798 transcription factors (some motifs shared due to high similarity), within the consensus set of open chromatin regions generated from bulk, sorted chromatin accessibility (ATAC-seq) data representing GABAergic neurons, glutamatergic neurons, oligodendrocytes and microglia[48]. To contextualize the footprinting findings within the *CD83* regulatory landscape, we computed enhancer-promoter links for *CD83* using the activity-by-contact model. This model integrated data from bulk, sorted chromatin accessibility (ATAC-seq)[48], and contact frequency matrices derived from bulk, sorted Hi−C data[48,102].

## Transverse colon proteomics

For LC–MS/MS, solubilized proteins were quantified (Thermo Fisher EZQ Protein Quantitation Kit or the Pierce BCA). Proteins were reduced with 50 mM dithiothreitol (Sigma-Aldrich) at 95 °C for 10 min and alkylated for 30 min with 40 mM iodoacetamide (Pierce). Proteins were digested using 2.0 μg of MS-grade porcine trypsin (Pierce), and peptides were recovered using S-trap Micro Columns (Protifi) per manufacturer directions. Recovered peptides were dried via speed vac and resuspended in 30 μl of 0.1% formic acid. All data-dependent mass spectra were collected in positive mode using an Orbitrap Fusion Lumos mass spectrometer (Thermo Scientific) coupled with an UltiMate 3000 UHPLC (Thermo Scientific). One μL of the peptide was fractionated using an Easy-Spray LC column (50 cm Å -75 μm ID, PepMap C18, 2 μm particles, 100 Å pore size, Thermo Scientific) with an upstream 300 μm Å -5 mm trap column. Electrospray potential was set to 1.6 kV and the ion transfer tube temperature to 300 °C. The mass spectra were collected using the "Universal" method optimized for peptide analysis provided by Thermo Scientific. Full MS scans (375–1500 m/z range) were acquired in profile mode with the following settings: Orbitrap resolution 120,000 (at 200 m/z), cycle time 3 s and mass range "Normal;" RF lens at 30% and the AGC set to "Standard"; maximum ion accumulation set to "Auto;" monoisotopic peak determination (MIPS) at "peptide" and included charge states 2–7; dynamic exclusion at 60 s, mass tolerance 10 ppm, intensity threshold at 5.0e3; MS/MS spectra acquired in a centroid mode using quadrupole isolation at 1.6 (m/z); collision-induced fragmentation (CID) energy at 35%, activation time 10 ms. Spectra were acquired over a 240-min gradient, flow rate 0.250 μl/min as follows: 0–3 min at 2%, 3–75 min at 2–15%, 75–180 min at 15–30%, 180–220 min at 30–35%, 220–225 min at 35–80% 225–230 at 80% and 230–240 at 80–5%.

## Label-free quantification (LFQ) and statistical analysis

Raw spectra were loaded into Proteome Discover 2.4 (Thermo Scientific), and protein abundances were determined using Uniprot (www.uniprot.org) Homo sapiens database (Hsap UP000005640.fasta). Protein abundances were determined using raw files and were searched using the following parameters: Trypsin as an enzyme, maximum missed cleavage site 3, min/max peptide length 6/144, precursor ion (MS1) mass tolerance at 20 ppm, fragment mass tolerance at 0.5 Da, and a minimum of 1 peptide identified. Carbamidomethyl (C) was specified as fixed modification and dynamic modifications set to Acetyl and Met-loss at the N-terminus, and oxidation of Met. A concatenated target/decoy strategy and a false-discovery rate (FDR) set to 1.0% were calculated using Percolator. Accurate mass and retention time of detected ions (features) using the Minora Feature Detector algorithm were then used to determine the area-under-the-curve (AUC) of the selected ion chromatograms of the aligned features across all runs and the relative abundances calculated.

**Table 2 | Clinical, neuropathological, and demographic information for the 433 subjects from the Fujita-ROSMAP cohort profiled by snRNA-seq**

| Fujita-ROSMAP | | AD | Control |
|---|---|---|---|
| Total subjects | | 270 | 163 |
| Expired age | Mean | 90.4 | 87.0 |
| | SD | 6.0 | 7.5 |
| Sex | F | 194 | 100 |
| | M | 76 | 63 |
| Race | White | 270 | 163 |
| PMI | Mean | 7.8 | 7.5 |
| | SD | 5.4 | 4.5 |
| Braak staging | 0 | | 6 |
| | 1 | | 24 |
| | 2 | 3 | 32 |
| | 3 | 61 | 67 |
| | 4 | 110 | 34 |
| | 5 | 93 | |
| | 6 | 3 | |
| CERAD | 1 (Definite) | 130 | |
| | 2 (Probable) | 139 | 12 |
| | 3 (Possible) | 1 | 42 |
| | 4 (No AD) | | 109 |
| NIA-Reagan | 1 (High) | 69 | |
| | 2 (Intermediate) | 201 | |
| | 3 (Low) | | 158 |
| | 4 (No AD) | | 5 |
| APOE | 22 | | 5 |
| | 23 | 34 | 25 |
| | 33 | 147 | 112 |
| | 24 | 6 | 1 |
| | 34 | 79 | 19 |
| | 44 | 4 | 1 |

*CERAD* semiquantitative measure of neuritic plaques, *NIA-R* NIA-Reagan diagnosis of AD.

**Table 3 | Clinical, neuropathological, and demographic information for the 427 subjects from the Mathys-ROSMAP cohort profiled by snRNA-seq**

| Mathys-ROSMAP | | AD | Control |
|---|---|---|---|
| Total subjects | | 238 | 189 |
| Expired age | Mean | 88.9 | 86.3 |
| | SD | 5.7 | 6.5 |
| Sex | F | 127 | 88 |
| | M | 111 | 101 |
| Race | White | 237 | 188 |
| | Black or African American | 1 | 1 |
| PMI | Mean | 7.6 | 7.7 |
| | SD | 5.3 | 7.0 |
| Braak staging | 0 | 1 | 7 |
| | 1 | 1 | 41 |
| | 2 | 6 | 59 |
| | 3 | 50 | 54 |
| | 4 | 66 | 28 |
| | 5 | 105 | |
| | 6 | 9 | |
| CERAD | 1 (Definite) | 126 | |
| | 2 (Probable) | 111 | 15 |
| | 3 (Possible) | 1 | 42 |
| | 4 (No AD) | | 132 |
| NIA-Reagan | 1 (High) | 83 | |
| | 2 (Intermediate) | 155 | |
| | 3 (Low) | | 183 |
| | 4 (No AD) | | 6 |
| APOE | 22 | | 3 |
| | 23 | 22 | 33 |
| | 33 | 124 | 128 |
| | 24 | 7 | 3 |
| | 34 | 76 | 21 |
| | 44 | 8 | |
| | NA | 1 | 1 |

*CERAD* semiquantitative measure of neuritic plaques, *NIA-R* NIA-Reagan diagnosis of AD.

## ROSMAP snRNA-seq processing and statistical analysis

The Religious Orders Study and Rush Memory and Aging Project (ROSMAP) are prospective cohort studies of aging and dementia[34]. Participants without known dementia agree to annual clinical evaluation and brain donation. Both studies were approved by an Institutional Review Board of Rush University Medical Center. All participants signed informed and repository consents, and an Anatomic Gift Act. Pathologic methods and APOE genotyping have been previously reported[103–105]. Clinical, postmortem neuropathological data, and demographics of the cohort profiled by the snRNA-seq study are reported in Tables 2 and 3.

For the Fujita-ROSMAP dataset, we downloaded all processed data (aligned UMI count matrices) generated from dorsolateral prefrontal cortex samples from the synapse (syn51123521)[27]. For each processed library batch, a Seurat object was created. Based on the cell annotation file (syn51218314), the object was demultiplexed into the objects for each individual. Eventually, all the demultiplexed objects from the same individual were merged into one object in Seurat. We applied the same analysis workflow as above in Seurat for each subject by filtering out low-quality nuclei and doublets, normalization, and clustering.

Integration of all the nuclei was accomplished in multiple steps. We first randomly split the 436 subjects into three batches. For each batch, reference-based integration was employed for the integration of all subjects in Seurat, using one male and one female as references.

RPCA was applied to the first 30 PCs of the 3000 variable genes to find the anchors before integration by SCTransform. The resulting three integrated Seurat objects were then integrated together using the atomic sketch integration method[106] by selecting and storing 50,000 representative cells ("atoms") from each dataset in the sampling step. We applied the same workflow as above for dimensionality deduction, clustering, and cell type annotation.

For DA population detection, the NIA-Reagan diagnosis[107] was used to dichotomize the subjects into AD and control. The same workflow was applied to identify the DA population in DAseq[21], except the values of k used for calculation of score vector with kNN were set 100–1000 with an increment of 100 to reduce the computing burden.

For the Mathys-ROSMAP dataset, we downloaded all processed raw data from prefrontal cortex samples from the synapse (syn52392369)[33] and converted it into a Seurat v5 subject. Following the Seurat v5 integration workflow, we split the dataset into 16 layers, one for each batch, to facilitate integration. The resulting Seurat objects were integrated together using the atomic sketch integration method by selecting and storing 50,000 representative cells ("atoms") from each dataset in the sampling step. We applied the same workflow as above for dimensionality deduction, clustering, and cell type annotation. Those cells with unknown cell types in both our and Mathy's annotations were filtered out.

For DA population detection, the NIA-Reagan diagnosis[107] was used to dichotomize the subjects into AD and control. Due to the large cell numbers, the same workflow was applied to identify the DA population in DAseq[21] in microglia cluster only, and the values of k used for calculation of score vector with kNN were set 100–1000 with an increment of 100 to reduce the computing burden. The threshold to get DA cells was set to (−0.5, 0.5).

In both datasets, *CR1* expression in oligodendrocytes was obtained by normalization in scran for all the nuclei within the cluster, followed by mean aggregation of the expression profiles across nuclei from each individual. Genotypes were obtained from WGS profiles (syn11707418). Association tests were performed using linear models in R, including age, gender, race, years of education, disease status, and PMI as covariates.

Antemortem blood measurements were obtained from Rush RADC Research Resource Sharing Hub (https://www.radc.rush.edu/requests.htm). We used the last available measurement within three years of death for each individual and performed linear regression of major allele dosage of rs679515 with the traits, including age at measurement, gender, and year of education as covariates. All AD-specific effects were modeled by adding an interaction term for each postmortem neuropathological measurement. Semiquantitative measurements (e.g., Braak stage or CERAD score) were treated as quantitative. Quantitative measurements (e.g., amyloid or tangles) were log-transformed.

### Reporting summary

Further information on research design is available in the Nature Portfolio Reporting Summary linked to this article.

## Data availability

All the raw and processed WGS, snRNA-seq, and metadata are available from Synapse with accession syn51753326. Source data are provided in this paper.

## Code availability

The code necessary to reproduce analyses is available at https://github.com/qwang178/Banner_snRNAseq_SFG. The current version is also available from Zenodo.

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

## Acknowledgements

The authors are grateful for the generous support of the NOMIS Foundation. B.R. is supported in part by NIA grants U01AG061835 and R21AG063068. Q.W. is supported in part by NIA grants U01AG061835. B.R. is grateful for the funding support received from The Benter Foundation. B.R. is grateful for funding support received from the Global Lyme Alliance. B.R. and D.M. are grateful for funding support received from the Arizona Alzheimer's Consortium. Q.W. and E.M.R. are grateful for additional funding from NIA grant P30AG072980. C.F. is supported by NIA grants 5U01AG046139-10 and R01AG062514. The Brain and Body Donation Program has been supported by the National Institute of Neurological Disorders and Stroke (U24 NS072026 National Brain and Tissue Resource for Parkinson's Disease and Related Disorders), the National Institute on Aging (P30AG19610 and P30AG072980, Arizona Alzheimer's Disease Center), the Arizona Department of Health Services (contract 211002, Arizona Alzheimer's Research Center), the Arizona Biomedical Research Commission (contracts 4001, 0011, 05-901, and 1001 to the Arizona Parkinson's Disease Consortium) and the Michael J. Fox Foundation for Parkinson's Research. We are grateful to the Banner Sun Health Research Institute Brain and Body Donation Program of Sun City, Arizona for the provision of human biological materials. BBDP resources can be requested at http://www.brainandbodydonationprogram.org. ROSMAP is supported by National Institute on Aging grants P30AG10161, P30AG72975, R01AG15819, R01AG17917, U01AG46152, and U01AG61356. ROSMAP resources can be requested at https://www.radc.rush.edu.

## Author contributions

E.M.R., W.L., D.M., K.V.K.J., J.T.D., B.R., T.G.B., and G.E.S designed the study. B.R. led the computational analyses. Q.W. and B.R. performed the computational analyses. Q.W. and B.R. wrote the paper. D.M. performed histochemical studies of CD83(+) microglia, generated transverse colon proteomics, and provided key interpretations of major findings within the study. T.G.B. and G.E.S. advised on subject and sample selection and provided biological samples for evaluation. T.G.B. and G.E.S. also performed a chart review of hematocrit values within the Banner study. K.V.K.J., W.L., J.A., E.A., and R.R. performed snRNA-seq data generation and data analysis. C.F. generated the transcriptional regulatory model for NR4A2. J.B. and P.R. performed the epigenomic landscape analysis of CD83 regulatory regions within microglia. T.L.K. assisted with the generation of proteomics data from the transverse colon. D.A.B. and P.L.D. provided data from ROSMAP and scientific inputs. All authors reviewed and edited the manuscript for important intellectual content.

## Competing interests

The authors declare no competing interests.
