## [Peer Review File · Nature Communications]

Single cell transcriptomes and multiscale networks from persons with and without Alzheimer's diseaseREVIEWER COMMENTS

Reviewer #1 (Remarks to the Author):

General assessment:

The manuscript by Wang and colleagues provides a resource of single nucleus RNA-seq dataset generated from the superior frontal gyrus (SFG) of the brain of donors from the Arizona Study of Aging and Neurodegenerative Disease. The authors establish the AD trait association of certain cell subtypes, perform a cell type specific eQTL analysis and identify peripheral traits (in blood and the gut) that correlate with AD associated changes in SFG. The strength of the manuscript is that no such resource from this well characterized cohort exists yet, the attempt to connect peripheral and CNS traits relevant to AD pathogenesis. The major weakness of the manuscript is that the analysis of the single nucleus RNA-sequencing experiment is extremely convoluted, with multiple layers of mapping, clustering and then re-merging the nuclei, with no clear explanation why these particular approaches were chosen, and not a more straightforward, conventional approach. Throughout this part of the manuscript there is no clear distinction between cell type and subset/sub-cluster, which makes it difficult to understand what exactly has been done and what exactly are the findings. Additionally, the formulation of some of the sentences make it very difficult to follow the snRNA-seq part of the manuscript (some examples are provided below in the specific comments). Accordingly, the manuscript would benefit from a major overhaul of the single nucleus RNA-sequencing part, including a thorough and critical reading by the senior authors. Separately, while the authors provided some orthogonal validation in a dataset that is in pre-print (<https://doi.org/10.1101/2022.11.07.515446>), for some reason they did not investigate how their data relates to findings of similar studies that have already been published (e.g. PMID: 31042697). Performing such comparison would help the readers in assessing the novelty of their findings. Additionally, a number of key points (discussed in detail below) need to be addressed for better reproducibility and interpretation of the data.

Specific comments:

- Line 81: "exceptionally 81 high-quality brain tissue" The authors should provide evidence for this claim. What were the criteria in assessing the quality of the brain tissue?
- Figure 1B: It is unclear what this panel represents. As per figure legend: "differentially abundant (DA) cell types observed in all major cell types". Presumably the authors mean that all major cell types had subsets that were differentially abundant between AD and control donors? Please specify.
- Line 92: "Unsupervised clustering by UMAP". The default clustering method in Seurat is KNN. UMAP is a visualization tool. Please specify the unsupervised hierarchical clustering approach used and rephrase. If indeed UMAP was used for clustering (which is non-conventional), please specify why.
- Line 533: Please specify the method used for "cell type label transfer".
- Line 565: Please specify what method was used to collapse the nuclei from the same individual in each cluster into one.
- Line 106: The authors state that "We then examined whether individual DA cell subpopulations are associated with clinical and neuropathological traits of interest to AD". But they also state that "The identified DA clusters were integrated with cell cluster annotation for their respective annotations." (line 561), and "The DA subpopulations were then collapsed into their respective cell type clusters and annotated as the DA cluster for each cell type" (line 103). So, really what they did was explored the association of the relative abundance of the identified clusters (which are at the cell type level) with AD traits, and not the association of Daseq derived sub-types with traits.
- Line 112: What do the authors mean by "considerable cell type heterogeneity across the spectrum of AD"? Please rephrase.
- Figure 2. "Differentially abundant nuclei vs non differentially abundant nuclei". How can nuclei be differentially abundant? They can belong to a differentially abundant cluster or sub-cluster, but then the figure legends should reflect that.
- Line 182: There is no causal inference testing approach described in reference #47.
- Supplementary figure 2b is difficult to assess with all the different lines representing Braak stages and the dots representing donors. Figure legend for section B lists only hematocrit, but there are 3 more measurements in this section. The number of donors used for each measurement

should be listed in the figure legends or on the figure. Statistical assessment of these associations should be provided.

- Supplementary tables need information about the content within the table itself.
- The finding that NR4A2 might be a key transcription factor regulating the CD83+ microglia subtype is intriguing – the authors should provide further support to this claim by an overexpression experiment in a suitable in vitro model system.
- The authors should provide a detailed analysis of how their subtypes relate to the subsets described by Mathys et al. (PMID: 31042697), including their relationship with clinicopathological traits shown for this dataset in Figure 1C

Reviewer #2 (Remarks to the Author):

The authors perform snRNAseq from SFG and WGS on 101 subjects from Banner Brain bank. Their main findings are 1) increased power to detect relevant DEGs with a differential abundance analysis, 2) the identification of an oligodendrocyte specific sn-eQTL signal for CR1 & hematological lab results and 3) identification of a differentially abundant microglial subtype associated that's also associated with IgG4 production in the transverse colon. Overall the study is well powered for the analyses they report, and has a reasonable replication strategy with the ROSMAP cohort. Furthermore, they apply standard and well accepted methods for their analyses so I have no major concerns on that front. That being said, there are a couple major and minor concerns that if addressed could strengthen the manuscript and make it higher impact to the broader AD research community.

Major Concerns

1. The authors argue that the differential abundance analysis increases power (by decreasing noise) compared to a standard DEG analysis, and use enrichment for GWAS genes as a proxy for improved power (and that the effect sizes are larger with DA v.s. DX analyses). The GWAS enrichment feels a little anecdotal, because I don't have a good intuition for why GWAS genes would be more enriched one way or another (they may just be glial genes and you see differential abundance of glia in AD). I would rather see a comparison of their DA DEGs with a meta-analysis of DX DEGs of their data with data reported in other major snRNA-seq AD studies (ROSMAP/SEA-AD/etc...), as a meta-analysis of the DX DEGs should reduce the variance of estimates and as such support the argument that the DA approach is more efficient more directly (e.g. if they see they get stronger concordance with the meta-analysis results and the DA results compared to the Banner alone DX DEGs). Otherwise I'd de-emphasize this result (e.g. put it in the supplement)
2. The AD specific eQTL interaction is super provocative for CR1, and it is unfortunate it does not replicate in ROSMAP. Have the authors considered testing for it in SEA-AD or another snRNA-seq data-set with WGS data? Also - even though it would be bulk tissue - it would be worth looking up the pQTLs for Banner and ROSMAP to see if that cell type specific QTL shows up in proteomic data? Those summary stats are publicly available I believe. It would strengthen the observation - particularly if the authors were willing to run the disease specific pQTL interaction analysis in proteomic data from ROSMAP or Banner.
3. I'm not sure what to make of the association with hematological traits - it would be nice to see the authors provide at least a rationale in the discussion as to why differential CR1 levels would lead to differences in RBC/Hb/hematocrit levels mechanistically - and more importantly why should the reader care? Are they proposing the hemotological parameters as a potential clinical biomarker for oligodendroglial CR1 activity in a tau dependent manner - or is it a clue as to the disease pathophysiological collapse of some key signaling pathway both in the periphery and in the CNS?
4. Their CD83+ subpopulation looks very similar to the senescent cluster 9 reported in Prater et al. (2023). Some discussion of if in fact this CD83+ population is senescent (or at an end stage of the disease process) is warranted - and comparison to the expression data of Prater et al. would significantly strengthen the manuscript and results around the CD83+ microglia.

Minor Concerns

1. Figure 1 legend labels are very small and hard to read. It would be nice to know what the take home message is from Sub-panel B in terms of the UMAP plot as far as differentially abundant cell types? It might be more illustrative to have a plot showing the # cells/how differentially abundant they are for the populations that are differentially "abundant" - I'm not sure what I should be taking away from where in the umap plot those cell pops are aside from the overall cell identity.

2. As far as figure 2, this is a very confusing plot. I think the authors are trying to make the point that focusing on DA DEGs decreases noise for DEG analysis? I would recommend simplifying the number of analysis facets to the main point the authors want to make - because currently they are showing differential expression, differential abundance, cell type, $-\log_{10}$ p-value of differential abundance, and a Fisher's exact test of enrichment with known GWAS genes - but only for differential abundance. Maybe split it into a table for the overlap test - and boxplots/violin plots for the \log_2 FC? I'm not sure the relationship between DA and DX is super relevant for the point they are trying to make - we know that there is gliosis so glial cell population increase and neurons are lost (which is the point of Figure 1 I think?)

3. Similarly figure 5 - is super busy, and I'm not sure the takeaway - I'd consider simplifying it to the main points - CD83+ microglia show interesting associations with disease (e.g. APOE and are DA) and peripheral measurables (IGHG4) - and potentially are in fact senescent? Up to the the authors to decide what point they want to really make clear to the readers.

Reviewer #3 (Remarks to the Author):

The investigation undertaken in this study sought to elucidate the augmentative impact of integrating single-cell RNA sequencing (scRNA-seq) data with diverse modalities on our understanding of Alzheimer's Disease (AD), with potential implications for translational applications. Specifically, the authors employed single-nucleus RNA sequencing (snRNA-seq) to analyze cortical tissue samples from 101 meticulously characterized subjects, amalgamating these data with whole genome sequences and diverse multiomics datasets, including genetics, proteomics, and clinical information, derived from the Banner Brain and Body Donation Program.

In the analysis of scRNA-seq data, the authors utilized the DAseq computational approach to identify differential abundance in cell subpopulations across major cell types. Focusing on over 9,000 cells from a total of 481,800 nuclei and 11 distinct clusters post-quality control filtering, the authors conducted differential analyses within major cell types, comparing differentially abundant (DA) subpopulation cells to non-DA cells, as well as standard differential analyses between AD and control conditions. Notably, AD risk genes were predominantly identified as significantly differentially expressed in the DA-only comparison. The integration of single-cell transcriptomics with whole genome sequencing aimed to delineate shared and specific expression quantitative trait loci (eQTLs) across cell types. The study also focused on the microglia subpopulation DA9, characterized by robust CD83 expression and identified as an AD-associated cluster. Analyses involved differential gene expression studies and the construction of a regulatory network.

The study's strengths are acknowledged; however, further clarification and detailed elucidation on certain aspects would enhance the overall understanding of the findings. We believe that providing additional insights and clarifications to our questions would significantly contribute to this paper.

Major Revision:

1. The provision of a synapse accession number by the authors is appreciated. Nonetheless, it is crucial that the authors make both the raw data and its accompanying metadata available for thorough examination by reviewers. Given the inherent sparsity of single-cell data, emphasizing the importance of rigorous quality assessment to ensure the reliability and integrity of downstream analyses is paramount.

2. The authors' contributions emphasize the significance of computational analyses in this study. To facilitate a comprehensive review, we recommend the inclusion of a code availability section,

detailing all necessary code for transparency and replicability.

3. The DA differential gene analyses and the identification of microglia DA9 subpopulations crucially depend on accurately delineating subpopulations associated with disease distinct from other cellular cohorts. Using Daseq, the authors identified 9,345 cells across 11 distinct DA subpopulations. In the supplementary table detailing DA distributions, DA1 to DA11 subpopulations range from 131 to 4505 cells. We suggest the authors undertake a more detailed examination of these DA subpopulations, extending beyond differential gene analyses between DA and non-DA populations, to assert the presence of AD-relevant cell subpopulations. Specifically, an exploration of why DA1 includes cells from various cell types, such as oligodendrocytes, excitatory neurons, inhibitory neurons, and astrocytes, and whether these cells share greater similarities amongst themselves than with cells in corresponding identified cell types, would enhance the interpretability of the findings. Moreover, an investigation into why these disparate cell-types cluster in DA1 is essential. Additionally, it's important to determine if the DA clustering uniquely identifies cells from a single individual, potentially selectively captured by the analytical program. An alternative strategy, where Daseq is executed on one cell type at a time, might yield a more refined identification of DA subpopulations within each cell type.

4. In addition to identifying DA subpopulations and their association with AD, the differential analysis conducted between DA cells and their non-DA counterparts, particularly in microglia, is of interest. For instance, in the microglia DA-only analyses, it is unclear whether the authors conducted differential analyses between DA and non-DA cells for all DA microglia or specifically within the DA9 subpopulation. The methodological framework indicates that genes detected in a minimum of 1% of cells in either condition were subjected to testing, suggesting that genes expressed in as few as four cells were considered. This raises concerns about the potential for false positives due to the minimal expression threshold applied.

5. The implementation of whole-genome sequencing and integration of genotype data with single-cell expression to establish cell-type-specific eQTLs is commendable. We propose the exploration of comprehensive eQTL colocalization analyses to maximize data utilization. This involves investigating whether an SNP is causally associated with both GWAS and eQTL signals within a specific locus, surpassing the current annotation methodology based on SNP identification in the GWAS catalog.

6. In examining eQTLs, the authors focused on CR1 in oligodendrocytes. Although CR1 is a recognized AD GWAS risk gene, and most cis-eQTLs in Figure 2 emanate from oligodendrocytes, the rationale behind emphasizing CR1, which is absent from the differential analyses in Figure 1E, requires clarification to enhance coherence and contextual understanding of the study's focus.

7. In Figure 4A, please clarify the genotype labels 0, 1, and 2 by adding actual genotype (A,T,G,C)

8. In Figure 4D, we request a modification of the color scheme and the inclusion of clear color legends to avoid confusion with the colors used for AD and Control in Figure 4A. Additionally, clarity on the interpretation of the p-values in Figure 4D is needed. Does the p-value indicate differences in multiple testing comparisons across various allele groups? The statistical significance of comparing group 0, with only two samples, is questionable, particularly when one sample's hematocrit score aligns with the mean or median scores of the other two groups.

9. The authors noted that microglia DA9 express CD83, supported by an image in Figure 5B showing microglia staining with CD83. However, Figure 5B does not conclusively demonstrate that cells expressing CD83 are exclusively associated with DA9, as this expression might not be unique to the DA9 subpopulation.

10. Regarding DA9 microglia, the observation of upregulated Disease-Associated Microglia (DAM) signatures prompts a request for the authors to provide DAM gene set scores across different microglial clusters.

11. Can the authors verify the accuracy of the information on line 323 (Figure 4A) to ensure alignment with the intended markings or annotations.

Point-by-Point Responses to Reviewer Comments

We would like to thank the reviewers for the very constructive comments and suggestions that were provided in the reviews of our previous submission. We are pleased to present a greatly restructured version of our study which has aimed to be maximally responsive to these thoughtful critiques.

Reviewer #1

The manuscript by Wang and colleagues provides a resource of single nucleus RNA-seq dataset generated from the superior frontal gyrus (SFG) of the brain of donors from the Arizona Study of Aging and Neurodegenerative Disease. The authors establish the AD trait association of certain cell subtypes, perform a cell type specific eQTL analysis and identify peripheral traits (in blood and the gut) that correlate with AD associated changes in SFG. The strength of the manuscript is that no such resource from this well characterized cohort exists yet, the attempt to connect peripheral and CNS traits relevant to AD pathogenesis.

Thanks to the reviewer for the positive summary of our study. We also appreciate the constructive critiques which we have considered carefully and which have substantially informed this revision.

The major weakness of the manuscript is that the analysis of the single nucleus RNA-sequencing experiment is extremely convoluted, with multiple layers of mapping, clustering and then re-merging the nuclei, with no clear explanation why these particular approaches were chosen, and not a more straightforward, conventional approach. Throughout this part of the manuscript there is no clear distinction between cell type and subset/sub-cluster, which makes it difficult to understand what exactly has been done and what exactly are the findings. Additionally, the formulation of some of the sentences make it very difficult to follow the snRNA-seq part of the manuscript (some examples are provided below in the specific comments). Accordingly, the manuscript would benefit from a major overhaul of the single nucleus RNA-sequencing part, including a thorough and critical reading by the senior authors.

- Line 106: *The authors state that “We then examined whether individual DA cell subpopulations are associated with clinical and neuropathological traits of interest to AD”. But they also state that “The identified DA clusters were integrated with cell cluster annotation for their respective annotations.” (line 561), and “The DA subpopulations were then collapsed into their respective cell type clusters and annotated as the DA cluster for each cell type” (line 103). So, really what they did was explored the association of the relative abundance of the identified clusters (which are at the cell type level) with AD traits, and not the association of DAseq derived sub-types with traits.*

- Line 565: *Please specify what method was used to collapse the nuclei from the same individual in each cluster into one.*

Thank you to the reviewer for this thoughtful feedback, which we have carefully considered and aimed to address in as thorough manner as possible. While we feel that some of the reviewer concerns can be assuaged through clarification of our language, we have also incorporated findings from an additional 427 snRNAseq samples, and performed extensive new analyses which we feel has strengthened our main findings (detailed below). In addition, we have performed a comprehensive and critical review of the manuscript to maximize readability. This includes harmonization of our language denoting cell populations, where we have restricted ourselves to the terms “cell subtype” and “cluster” which we use synonymously, but which we vary in use depending on grammatical context.

Regarding the reviewer concern of “multiple layers of mapping, clustering and then re-merging the nuclei, with no clear explanation why these particular approaches were chosen, and not a more straightforward, conventional approach” we think that part of this impression was due to unclear writing on our part, where our description incorrectly implied that we were combining nuclei profiles for an individual within a cell subtype to perform the clinicopathological association testing; whereas we were actually just looking at the traits of the unique set of subjects that participated in a particular cell cluster. These results are summarized in the new **Figure 1C**, which includes an updated caption and associated results and methods description.

Separately, while the authors provided some orthogonal validation in a dataset that is in pre-print (<https://doi.org/10.1101/2022.11.07.515446>), for some reason they did not investigate how their data relates to findings of similar studies that have already been published (e.g. PMID: 31042697). Performing such comparison would help the readers in assessing the novelty of their findings. Additionally, a number of key points (discussed in detail below) need to be addressed for better reproducibility and interpretation of the data.

- The authors should provide a detailed analysis of how their subtypes relate to the subsets described by Mathys et al. (PMID: 31042697), including their relationship with clinicopathological traits shown for this dataset in Figure 1C

We completely agree with the reviewer on the importance of comparing our observations with those presented in the published scientific literature. We were initially limited in our ability to perform this comparison with the Mathys 2019 data set. Like many in the field, we were strongly influenced by this seminal paper, however the limitations in sample size (n=48 samples) meant that we were unable to apply the differential abundance (DA) approach that we had used for our own data, which limited our ability to perform a straightforward comparison of our DA cell clustering results.

We are pleased to report a comparison of our observations against the much larger, updated Mathys et al 2023 (PMID: 37774677) data set, which comprises 427 snRNAseq samples. These samples derive from the ROSMAP cohort, including many samples not profiled within the 424 samples included in the Fujita et al preprint that we had previously incorporated. We now refer to these data sets as Mathys-ROSMAP and Fujita-ROSMAP throughout our manuscript. Both the Mathys-ROSMAP and the Fujita-ROSMAP data sets are large enough to apply the DA approach we used on our own Banner data, and we have included these comparisons within the manuscript.

We are encouraged to note the presence of a comparable CD83(+) microglia cluster within both additional cohorts (See **Supplementary Figure 4**), as well as a replication of our previous observation that CD83(+) microglia are associated with an increased neuritic plaque burden in AD subjects (**Supplementary Figure 5**). In addition, we are also pleased to report an additional replication of our CR1 / rs679515 eQTL association in the newly incorporated Mathys-ROSMAP data (**Supplementary Figure 2**) with an accompanying colocalization analysis (**Figure 3B**) which supports a shared genetic basis for the variant that alters oligodendrocyte CR1 expression and AD risk. We also report a replication of an AD-specific interaction of this eQTL which we had previously observed in our own Banner data, but not in the Fujita-ROSMAP data. As described in our updated manuscript, we now observe a significant interaction of this association with neuritic amyloid plaque density (**Supplementary Table 5**), which to our knowledge is previously unreported in the scientific literature.

Specific comments:

- Line 81: “exceptionally 81 high-quality brain tissue” The authors should provide evidence for this claim. What were the criteria in assessing the quality of the brain tissue?

Thank you for the opportunity to clarify this statement; here we are referring to the very low median post-mortem interval (PMI) of 3 hours for the donors within the Banner Brain & Body Donation Program. We have added this additional detail to the manuscript text to justify this statement.

- Figure 1B: It is unclear what this panel represents. As per figure legend: “differentially abundant (DA) cell types observed in all major cell types”. Presumably the authors mean that all major cell types had subsets that were differentially abundant between AD and control donors? Please specify.

The reviewer’s interpretation is correct; here we are showing that there are DA cell clusters identified within each of the six major cell types. We have updated the **Figure 1B** caption to clarify this.

- Line 92: “Unsupervised clustering by UMAP”. The default clustering method in Seurat is KNN. UMAP is a visualization tool. Please specify the unsupervised hierarchical clustering approach used and rephrase. If indeed UMAP was used for clustering (which is non-conventional), please specify why.

Thank you for the opportunity to clarify this. The reviewer is indeed correct that we adopted the standard KNN approach for clustering, and then used UMAP for visualization only. We have updated the Figure legend to reflect this.

- Line 533: Please specify the method used for “cell type label transfer”.

Thank you for the opportunity to clarify this. We can confirm that we used the “FindTransferAnchors” and “TransferData” functions in Seurat. We have updated the Method section of our manuscript with this additional detail.

- Line 112: What do the authors mean by “considerable cell type heterogeneity across the spectrum of AD”? Please rephrase.

Thank you for the opportunity to clarify this. Here we were referring to the changes in cell type fraction that occur at different stages throughout the course of Alzheimer’s disease.

We have rewritten this sentence to the following: “Furthermore, the differential association of DA subpopulations with varying stages of neuropathological severity is consistent with the considerable changes in cell type fraction that are observed at different stages of AD.”

- Figure 2. “Differentially abundant nuclei vs non differentially abundant nuclei”. How can nuclei be differentially abundant? They can belong to a differentially abundant cluster or sub-cluster, but then the figure legends should reflect that.

We have rewritten the Figure 2 (now **Supplementary Figure 1**) legend to better communication this important distinction: “DA: Differentially abundant vs non differentially abundant nuclei clusters; DX: AD nuclei vs age-matched control nuclei clusters”

- Line 182: There is no causal inference testing approach described in reference #47.

Thank you for altering us to this mis citation, which we have corrected to indicate the inference testing methodology and software (respectively):

Millstein, Joshua, Bin Zhang, Jun Zhu, and Eric E. Schadt. "Disentangling molecular relationships with a causal inference test." *BMC genetics* 10 (2009): 1-15.

Millstein, Joshua, Gary K. Chen, and Carrie V. Breton. "Cit: hypothesis testing software for mediation analysis in genomic applications." *Bioinformatics* 32, no. 15 (2016): 2364-2365.

Supplementary figure 2b is difficult to assess with all the different lines representing Braak stages and the dots representing donors. Figure legend for section B lists only hematocrit, but there are 3 more measurements in this section. The number of donors used for each measurement should be listed in the figure legends or on the figure. Statistical assessment of these associations should be provided.

Thank you for noting this. We have revised **Supplementary Figure 2B** to include sample numbers and the interaction term P-values for each of the four traits noted (Hematocrit, Hemoglobin, Red Blood Cell Count and Anemia). We have also updated the figure description in the results text to point to **Supplementary Table 5**, which includes the full results of the association testing. We have updated the caption accordingly:

“Supplementary Figure 2: Relationship between rs679515 genotype and *CR1* gene expression in oligodendrocytes and antemortem hematological measurements in the ROSMAP cohorts. **(A)** CR1 expression stratified by AD diagnosis (Diagnosis was dichotomized by NIA-Reagan score). **(B)** Hematological parameter associations with rs679515 genotype, stratified by Braak score. Pvalues for statistical interaction between rs679515 dosage and Braak score shown. Sample number (n) shown. Detailed statistical metrics for each model are reported in Supplementary Table 5”.

- *Supplementary tables need information about the content within the table itself.*

We agree with the reviewer, and have added a README tab at the beginning of each supplementary table.

- *The finding that NR4A2 might be a key transcription factor regulating the CD83+ microglia subtype is intriguing – the authors should provide further support to this claim by an overexpression experiment in a suitable in vitro model system.*

We concur with the reviewer on the importance of presenting stronger evidence for the regulatory connection between CD83 and its predicted enhancer with bound NR4A2. To that end, we analyzed the chromatin accessibility correlation between this enhancer and the CD83 promoter (**panel A** below), utilizing 107 FANS-sorted ATAC-seq samples from the postmortem brains of Alzheimer's disease cases and controls (Kosoy et al, 2022). This is the same dataset we initially used for assessing enhancer-promoter interactions and transcription factor binding. Our analysis revealed a significant correlation (Spearman $\rho=0.44$, P-value=1.9E-5; Pearson R=0.47, P-value=4.6E-7; **panel B** below), which supports the proposed regulatory influence of this enhancer on CD83. We have added this analysis to the Results section of our manuscript.

Reviewer #2 (Remarks to the Author):

The authors perform snRNAseq from SFG and WGS on 101 subjects from Banner Brain bank. Their main findings are 1) increased power to detect relevant DEGs with a differential abundance analysis, 2) the identification of an oligodendrocyte specific sn-eQTL signal for CR1 & hematological lab results and 3) identification of a differentially abundant microglial subtype associated that's also associated with IgG4 production in the transverse colon. Overall the study is well powered for the analyses they report, and has a reasonable replication strategy with the ROSMAP cohort. Furthermore, they apply standard and well accepted methods for their analyses so I have no major concerns on that front. That being said, there are a couple major and minor concerns that if addressed could strengthen the manuscript and make it higher impact to the broader AD research community.

Thanks to the reviewer for these thoughtful critiques and suggestions.

The authors argue that the differential abundance analysis increases power (by decreasing noise) compared to a standard DEG analysis, and use enrichment for GWAS genes as a proxy for improved power (and that the effect sizes are larger with DA v.s. DX analyses). The GWAS enrichment feels a little anecdotal, because I don't have a good intuition for why GWAS genes would be more enriched one way or another (they may just be glial genes and you see differential abundance of glia in AD). I would rather see a comparison of their DA DEGs with a meta-analysis of DX DEGs of their data with data reported in other major snRNA-seq AD studies (ROSMAP/SEA-AD/etc...), as a meta-analysis of the DX DEGs should reduce the variance of estimates and as such support the argument that the DA approach is more efficient more directly (e.g. if they see they get stronger concordance with the meta-analysis results and the DA results compared to the Banner alone DX DEGs). Otherwise I'd de-emphasize this result (e.g. put it in the supplement)

and

As far as figure 2, this is a very confusing plot. I think the authors are trying to make the point that focusing on DA DEGs decreases noise for DEG analysis? I would recommend simplifying the number of analysis facets to the main point the authors want to make - because currently they are showing differential expression, differential abundance, cell type, $-\log_{10}$ p-value of differential abundance, and a Fisher's exact test of enrichment with known GWAS genes - but only for differential abundance. Maybe split it into a table for the overlap test - and boxplots/violin plots for the \log_2 FC? I'm not sure the relationship between DA and DX is super relevant for the point they are trying to make - we know that there is gliosis so glial cell population increase and neurons are lost (which is the point of Figure 1 I think?)

We agree with the reviewer; the key message of Figure 2 may not be worth the effort required to understand it given the other findings of the manuscript, so we have removed it as **Figure 2** and moved it to **Supplementary Figure 1**. Our goal with this analysis was to establish a justification for using a DA approach compared with conventional DEG approach, using any differential enrichment for AD GWAS genetics as a proxy for informative biological networks that could be used as inputs into downstream analyses. We don't necessarily think these enrichments are explained away by changes in cell-type fractions since each enrichment analysis is performed within each specific cell type (subsetting the enrichment background to genes expressed in that cell type), so think it still has some marginal value in justifying our analytical choices, though have reduced the emphasis. We also added a complementary MAGMA analysis which was supportive of an increased enrichment among microglia DA genes compared with DX genes (**Supplementary Table 2**).

The AD specific eQTL interaction is super provocative for CR1, and it is unfortunate it does not replicate in ROSMAP. Have the authors considered testing for it in SEA-AD or another snRNA-seq data-set with WGS data? Also - even though it would be bulk tissue - it would be worth looking up the pQTLs for Banner and ROSMAP to see if that cell type specific QTL shows up in proteomic data? Those summary stats are publicly available I believe. It would strengthen the observation - particularly if the authors were willing to run the disease specific pQTL interaction analysis in proteomic data from ROSMAP or Banner.

Thank you to the reviewer for these suggestions. We're pleased to share several extensions of these analyses, which have deepened our understanding of these findings. We have now included a comparison of our observations against the much larger, updated Mathys et al 2023 (PMID: 37774677) data set, which comprises 427 snRNA-seq samples from dorsolateral prefrontal cortex. These samples derive from the ROSMAP cohort, including many samples not profiled within the 424 samples included in the Fujita et al preprint that we had previously incorporated. We now refer to these data sets as Mathys-ROSMAP and Fujita-ROSMAP throughout our manuscript.

We were encouraged to observe an additional replication of our CR1 / rs679515 eQTL association in the newly incorporated Mathys-ROSMAP data with an accompanying colocalization analysis (updated **Figure 3B**). In brief, we demonstrate a posterior probability of 0.985 that a single causal variant, located within the CR1 eQTL region (250kb either side of the CR1 gene body) explains both the Alzheimer's disease GWAS risk signal, and the oligodendrocyte CR1 eQTL signal contained within that region. Further, we observe a posterior probability of 0.82 that rs679515 is the causal variant that drives both traits. We also report a replication

of an AD-specific interaction of this eQTL which we had previously observed in our own Banner data, but not in the Fujita-ROSMAP data. As described in our updated manuscript, we now observe a significant

interaction of this association with neuritic amyloid plaque density (**Supplementary Table 5**), so it appears that some kind of disease-interaction with CR1 expression at this locus may be a robust observation.

We also took the reviewer's advice to inspect potential pQTL associations for CR1 / rs679515, particularly the AMP-AD data that underpinned the Wingo *et al* 2023 pQTL study (PMID: 37653343), but were interested to note that CR1 was not detected at sufficient abundance in any of regions / cohorts (admittedly at the bulk tissue level) to be included within the scope of the pQTL analyses. This interesting question may require a cell-level proteomics resource to properly resolve. We have included a new reference to a plasma pQTL relationship between CR1 / rs679515 detected in plasma (Pietzner 2021, PMID: 34648354), which would seem to raise the credibility of a pQTL association also existing within oligodendrocytes.

I'm not sure what to make of the association with hematological traits - it would be nice to see the authors provide at least a rationale in the discussion as to why differential CR1 levels would lead to differences in RBC/Hb/hematocrit levels mechanistically - and more importantly why should the reader care? Are they proposing the hemotological parameters as a potential clinical biomarker for oligodendroglial CR1 activity in a tau dependent manner - or is it a clue as to the disease pathophysiological collapse of some key signaling pathway both in the periphery and in the CNS?

Thanks to the reviewer for raising this intriguing series of questions. We're not clear yet whether our combined observations represent some kind of crosstalk between oligodendrocytes and hematological tissues, or are phenomena that share conserved molecular networks but which are not necessarily linked mechanistically beyond the genetic perturbation on CR1 that rs679515 exerts on both systems. Given recent reports that rs679515 is a pQTL for CR1 protein abundance in plasma (Pietzner, 2021), it does perhaps seem more plausible that this is a robust, multi-tissue QTL relationship manifesting in disparate tissues in an independent manner, without needing to necessarily invoke a unified mechanism that links the two. In this scenario, whether the AD-relevant pathomechanism(s) are instantiated within oligodendrocytes, hematological cells, both, or as yet undiscovered tissues is not certain, though plausible hypotheses exist for how CR1 dysregulation in each of these tissues might impact AD. We do note our newly added finding of colocalization of the genetic signal for CR1 expression in oligodendrocytes with the AD GWAS signal to rs679515, as well as the emergence of an amyloid-specific interaction between rs679515 and CR1 within the Mathys-ROSMAP data, which offers new approaches for illuminating whether CR1 dysregulation in oligodendrocytes might contribute to AD. Tissue-specific CR1 perturbation mouse models may also help to dissect this.

We have expanded our discussion around the interpretation of these findings as follows:

“CR1 has been associated with AD via several polymorphisms⁴⁷⁻⁵⁰ though findings have been variable across datasets⁵¹ and clarifying whether the pathogenic mechanism occurs in the central nervous system (CNS) or periphery remains elusive. CR1 encodes a transmembrane glycoprotein receptor for multiple ligands (including complement components C3b and C4b) and is abundantly expressed on erythrocytes and leukocytes⁵², though also on tissue-resident macrophages and dendritic cells and on multiple brain cell types, including microglia, astrocytes⁵³, and oligodendrocytes⁵⁴. Peripherally, C3b-opsonized immune complexes (including amyloid aggregates) bind to CR1 on erythrocytes and undergo eventual phagocytic clearance in the liver and spleen⁵², representing a major hypothesized mechanism that links CR1 genetic variants with AD risk^{55,56}. Despite this notion, plasma CR1 levels have not been reported to be altered in AD, though are modestly increased in carriers of several CR1-linked AD GWAS risk variants⁵⁷. We observed the rs679515 / CR1 eQTL association only in oligodendrocytes, with network findings that were sufficiently reminiscent of canonical erythrocyte regulators to motivate our investigation of effects upon peripheral hematocrit. We also note recent reports that rs679515 is a protein QTL for plasma CR1

abundance⁵⁸. Overall, our findings align with a potentially multi-tissue impact of rs679515 / *CR1*, although whether the oligodendrocyte rs679515 / *CR1* network actually reflects a key AD pathomechanism or is just conserved enough as to suggest the relevant peripheral mechanism remains unresolved. The effects of rs679515 on both hematological traits and oligodendrocytes is consistent with a robust genetic impact upon *CR1* abundance across diverse tissues and need not necessarily imply a unified cross-tissue disease pathomechanism. However, the GWAS loci for *CR1* have also been independently found to be co-localized in oligodendrocytes in AD, by an integrative analysis of multiple snRNA-seq datasets from several different brain regions²⁹. We also note recent reports by Yang et al⁵⁹ on AD-relevant activity of *CR1* within perivascular macrophages (PVM). Given complex reciprocal interactions between oligodendrocytes and the neurovascular unit^{60,61}, simultaneous evaluation of *CR1* and AD GWAS loci in PVM and oligodendrocytes may illuminate a shared mechanism that could explain the observed CNS as well as hematological effects in a parsimonious manner. Further study of whether peripheral *CR1* levels associate with rs679515 in an AD-specific manner may also be an informative direction for future investigation.”

Their CD83+ subpopulation looks very similar to the senescent cluster 9 reported in Prater et al. (2023). Some discussion of if in fact this CD83+ population is senescent (or at an end stage of the disease process) is warranted - and comparison to the expression data of Prater et al. would significantly strengthen the manuscript and results around the CD83+ microglia.

Thank you for referring us to this valuable study and resource. We performed a comparison of our CD83(+) microglia (referenced against nonCD83(+) microglia) upregulated DEG against those from all 9 clusters reported within the Prater et al study using a hypergeometric test, using a gene background of the union of the genes detected in >1% of our microglia and those present as DEGs in all 9 clusters. We observed striking enrichments against several Prater clusters, most significantly against Cluster 5 (Stress, autophagy and endolysosomal network, interestingly, also including CD83) and also Cluster 9 (Senescent-like, and immediately downstream of Cluster 5) as the reviewer suggested. The only clusters which we did not observe some nominal enrichment was Cluster 2 (Homeostatic) and Cluster 10 (Cell-cycle).

Prater Cluster	# DEG	# overlap	P	OR
2	10	1	1.95E-01	5.09
3	176	41	5.99E-31	15.55
4	277	33	1.07E-15	6.71
5	322	197	3.51E-267	158.82
6	954	87	5.75E-32	5.74
7	263	78	5.34E-68	24.36
8	287	80	2.17E-67	22.45
9	393	153	3.46E-161	50.12
10	661	18	2.72E-01	1.29

We have expanded our discussion around the interpretation of these findings as follows:

“We then compared CD83(+) microglia transcriptomic signatures against a recent snRNAseq study of human microglia in AD (**Supplementary Table 7**). Prater et al applied a trajectory mapping approach to microglia-enriched snRNA-seq generated from 22 postmortem DLPFC samples and identified 10 microglial clusters with distinct transcriptomic, molecular pathway, and phenotypic profiles. We observed

striking enrichments against two microglial clusters in particular: Cluster 5 (which contains CD83 and is primarily enriched for stress-autophagy and endolysosomal network pathway activity, P-value: 7.13e-228, Odds Ratio: 98.84) and Cluster 9 (primarily enriched for senescence, iron homeostasis and cytokine production pathway activity, P-value: 3.02e-131, Odds Ratio: 31). In combination with the direct enrichment of CD83(+) microglia drivers indicating perturbed iron processing (**Figure 4E**), this may suggest that our reported CD83(+) microglia are actually a senescent type of microglia”

Minor Concerns

Figure 1 legend labels are very small and hard to read. It would be nice to know what the take home message is from Sub-panel B in terms of the UMAP plot as far as differentially abundant cell types? It might be more illustrative to have a plot showing the # cells/how differentially abundant they are for the populations that are differentially "abundant" - I'm no sure what I should be taking away from where in the umap plot those cell pops are aside from the overall cell identity.

Thank you for these suggestions. We have substantially redesigned **Figure 1** to more fully communicate the experimental workflow and eventual differential abundance population detection. We have also included a table (**Figure 1C**) as suggested by the reviewer, and a revised version of our clinicopathological trait association heatmap.

Similarly figure 5 - is super busy, and I'm not sure the takeaway - I'd consider simplifying it to the main points - CD83+ microglia show interesting associations with disease (e.g. APOE and are DA) and peripheral measurables (IGHG4) - and potentially are in fact senescent? Up to the the authors to decide what point they want to really make clear to the readers.

We agree with the reviewer. We have simplified the layout of Figure 5 (now **Figure 4**), moving the NR4A2 transcription factor findings into **Supplementary Figure 6** and instead focused on the main findings around CD83(+) microglia.

Reviewer #3 (Remarks to the Author):

The investigation undertaken in this study sought to elucidate the augmentative impact of integrating single-cell RNA sequencing (scRNA-seq) data with diverse modalities on our understanding of Alzheimer's Disease (AD), with potential implications for translational applications. Specifically, the authors employed single-nucleus RNA sequencing (snRNA-seq) to analyze cortical tissue samples from 101 meticulously characterized subjects, amalgamating these data with whole genome sequences and diverse multiomics datasets, including genetics, proteomics, and clinical information, derived from the Banner Brain and Body Donation Program.

In the analysis of scRNA-seq data, the authors utilized the Daseq computational approach to identify differential abundance in cell subpopulations across major cell types. Focusing on over 9,000 cells from a total of 481,800 nuclei and 11 distinct clusters post-quality control filtering, the authors conducted differential analyses within major cell types, comparing differentially abundant (DA) subpopulation cells to non-DA cells, as well as standard differential analyses between AD and control conditions. Notably, AD risk genes were predominantly identified as significantly differentially expressed in the DA-only comparison. The integration of single-cell transcriptomics with whole genome sequencing aimed to delineate shared and specific expression quantitative trait loci (eQTLs) across cell types. The study also focused on the microglia subpopulation DA9, characterized by robust CD83 expression and identified as an AD-associated cluster. Analyses involved differential gene expression studies and the construction of a regulatory network.

The study's strengths are acknowledged; however, further clarification and detailed elucidation on certain aspects

would enhance the overall understanding of the findings. We believe that providing additional insights and clarifications to our questions would significantly contribute to this paper.

Thanks to the reviewer for the detailed and positive summary of our study. We also appreciate the constructive critiques which we considered carefully and have substantially informed this revision.

The provision of a synapse accession number by the authors is appreciated. Nonetheless, it is crucial that the authors make both the raw data and its accompanying metadata available for thorough examination by reviewers. Given the inherent sparsity of single-cell data, emphasizing the importance of rigorous quality assessment to ensure the reliability and integrity of downstream analyses is paramount.

The authors' contributions emphasize the significance of computational analyses in this study. To facilitate a comprehensive review, we recommend the inclusion of a code availability section, detailing all necessary code for transparency and replicability.

Thank you for raising this important consideration. We have deposited all code necessary to reproduce our key analyses, which can be found at: https://github.com/qwang178/Banner_snRNAseq_SFG

We have updated the manuscript with a code availability section also:

“Code Availability

All code necessary to reproduce these analyses are available at https://github.com/qwang178/Banner_snRNAseq_SFG.”

3. The DA differential gene analyses and the identification of microglia DA9 subpopulations crucially depend on accurately delineating subpopulations associated with disease distinct from other cellular cohorts. Using DAseq, the authors identified 9,345 cells across 11 distinct DA subpopulations. In the supplementary table detailing DA distributions, DA1 to DA11 subpopulations range from 131 to 4505 cells. We suggest the authors undertake a more detailed examination of these DA subpopulations, extending beyond differential gene analyses between DA and non-DA populations, to assert the presence of AD-relevant cell subpopulations. Specifically, an exploration of why DA1 includes cells from various cell types, such as oligodendrocytes, excitatory neurons, inhibitory neurons, and astrocytes, and whether these cells share greater similarities amongst themselves than with cells in corresponding identified cell types, would enhance the interpretability of the findings. Moreover, an investigation into why these disparate cell-types cluster in DA1 is essential. Additionally, it's important to determine if the DA clustering uniquely identifies cells from a single individual, potentially selectively captured by the analytical program. An alternative strategy, where DAseq is executed on one cell type at a time, might yield a more refined identification of DA subpopulations within each cell type.

Thank you for pointing out this apparent disparity. We have since carefully evaluated the composition of DA1, which is the largest DA population in our study. We can confirm that the vast majority of the cells within this population are from one single cell type, i.e. neurons. We identified only one astrocyte, one microglia, one oligodendrocyte, and nineteen oligodendrocyte precursor cells within this population (See **Supplementary Table 1**), which totals less than 0.5% of the total cell count within DA1. These anomalous cells were also excluded in our downstream analysis (i.e. not counting towards DA1). A mixture of cell annotation for a very small fraction of cells may be expected for single cell transcriptomics, likely originating from a minority of lower quality cells that may be missing marker gene reads and thus leading to misclassification. We also confirm that, for each DA population, cells were sampled from large numbers of unique subjects, ranging from 22 to 66 as reported in **Supplementary Table 1**. We also reran DAseq restricting our analysis only to the microglia cell type, and using the same parameters), and we were able

to identify 260 out of the 352 DA cells (CD83+ microglia). We acknowledge that the identification of DA cells relies on the choice of a few parameters of DAseq, similar as those used in DEG analysis (ie. logFC cutoff). For this study we aimed to apply a reasonable, global approach to systematically analyze all available DA populations in all cell types profiled. Despite this, we do appreciate the reviewer's suggestion of performing within-cell-type DA identification and will be including it as part of an analytic plan for further snRNAseq we are generating from additional brain regions and which we will be reporting in a future study.

4. In addition to identifying DA subpopulations and their association with AD, the differential analysis conducted between DA cells and their non-DA counterparts, particularly in microglia, is of interest. For instance, in the microglia DA-only analyses, it is unclear whether the authors conducted differential analyses between DA and non-DA cells for all DA microglia or specifically within the DA9 subpopulation. The methodological framework indicates that genes detected in a minimum of 1% of cells in either condition were subjected to testing, suggesting that genes expressed in as few as four cells were considered. This raises concerns about the potential for false positives due to the minimal expression threshold applied.

We apologize for the confusion caused by the language denoting cell populations in our manuscript, which since has been updated. Our DEG analysis was performed within each major cell type, in particular a comparison between DA vs nonDA cells. The DA9 subpopulation just refers to the DA population that we detected in microglia (DA9 was the only DA population we detected in microglia, but 9th in the whole dataset when considering all cell types). In the DEG analysis, min.pct=0.01 is the default value set in Seurat FindMarker function. We have also reported the percent expressed in the two conditions in **Supplementary Table 2**. In our case, since the nonDA population for each of the cell types consists of >10k cells, then even at a minimal 1% threshold, we can assume that a gene is present in at least 100 cells, of that cell-type across the full data set, which in encouraging that any declared DEGs won't be based on observations from just a few cells.

5. The implementation of whole-genome sequencing and integration of genotype data with single-cell expression to establish cell-type-specific eQTLs is commendable. We propose the exploration of comprehensive eQTL colocalization analyses to maximize data utilization. This involves investigating whether an SNP is causally associated with both GWAS and eQTL signals within a specific locus, surpassing the current annotation methodology based on SNP identification in the GWAS catalog.

Thanks to the reviewer for making this thoughtful suggestion to incorporate colocalization analysis, which we adopted and now included within an updated **Figure 3B** and **Supplementary Table 4**. In brief, we are now able to demonstrate a posterior probability of 0.985 that a single causal variant, located within the CR1

eQTL region (250kb either side of the CR1 gene body) explains both the Alzheimer's disease GWAS risk signal, and the oligodendrocyte CR1 eQTL signal contained within that region. Further, we observe a posterior probability of 0.817 that rs679515 is the causal variant that drives both traits.

6. In examining eQTLs, the authors focused on CR1 in oligodendrocytes. Although CR1 is a recognized AD GWAS risk gene, and most cis-eQTLs in Figure 2 emanate from oligodendrocytes, the rationale behind emphasizing CR1, which is absent from the differential analyses in Figure 1E, requires clarification to enhance coherence and contextual understanding of the study's focus.

Thanks to the reviewer for the opportunity to clarify this thread of reasoning. As the reviewer

notes, this aspect of our analyses proceeded as an effort to annotate the variants that emerged in our cell-type cis-eQTL analysis, whereby several of these variants emerged as AD risk loci (as indexed through GWAS catalog). This AD-association thus formed the motivation for us to then consider the cell type (oligodendrocytes) and the gene (CR1) that had implicated these variants, and then further motivated the subsequent CR1/oligodendrocyte network analysis, which implicated the hematological trait assessments etc. The recently incorporated colocalization analysis (precipitated by the reviewer) also now forms part of the focus on CR1.

We have updated the results section of our manuscript to better delineate this motivation:

“We observed that reported two AD risk loci, rs1111832831 (chr1:207677194) and rs94297802 (chr1:1:207493845) are both eQTLs for Complement Receptor 1 (CR1) expression in oligodendrocytes. Recently, Fujita et al and Mathys et al published analyses based on snRNA-seq from 433 and 427 dorsolateral prefrontal cortex samples respectively, from subjects included within the Religious Orders Study and Rush Memory and Aging Project (ROSMAP) cohort. Using these data (Fujita-ROSMAP and Mathys-ROSMAP DLPFC snRNA-seq) we confirmed the association between both loci and CR1 expression in oligodendrocytes (**Supplementary Table 4**). Fujita et al also reported the nearby most significant AD risk locus rs679515 (chr1:207577223, Major allele: C, Minor allele: T, Reference allele: T) as an eQTL in oligodendrocytes, although this variant had not emerged in our analysis. Examination of our eQTL results revealed that rs679515 had been removed following linkage disequilibrium-based pruning with rs9429780. We then confirmed that rs679515 was also an eQTL for CR1 in oligodendrocytes in our dataset by a direct association test, in agreement with Fujita et al (**Supplementary Table 4**), where minor allele T (also the AD risk allele) is associated with higher CR1 expression. Consistent with Fujita et al, we also observed colocalization of the genetic signal driving CR1 expression in oligodendrocytes with AD GWAS risk signal within the Banner cohort (posterior probability: 0.985), with rs679515 emerging as the most likely causal variant explaining this shared signal (posterior probability: 0.817, **Figure 3B**, **Supplementary Table 4**). Further examination of the association between rs679515 and oligodendrocyte expression of CR1 within the Banner SFG data revealed that this association was driven by the association within AD subjects (**Figure 3A**, **Supplementary Table 4**), indicating an AD-specific interaction (AD

diagnosis/dosage interaction p value = $2.1e-5$) between allele dosage and CR1 expression within the SFG samples. We did not observe a significant AD-specific interaction within the Fujita-ROSMAP or Mathys-ROSMAP data, (**Supplementary Figure 2A, Supplementary Table 5**), however we did observe an interaction with amyloid plaque density within the Mathys-ROSMAP DLPFC cohort (P -value $<2.96e-2$, **Supplementary Table 5**).”

7. *In Figure 4A, please clarify the genotype labels 0, 1, and 2 by adding actual genotype (A,T,G,C)*

Thanks to the reviewer for the opportunity to clarify this. We have replaced the convention of showing allele dosage with genotype.

8. *In Figure 4D, we request a modification of the color scheme and the inclusion of clear color legends to avoid confusion with the colors used for AD and Control in Figure 4A. Additionally, clarity on the interpretation of the p-values in Figure 4D is needed. Does the p-value indicate differences in multiple testing comparisons across various allele groups? The statistical significance of comparing group 0, with only two samples, is questionable, particularly when one sample's hematocrit score aligns with the mean or median scores of the other two groups.*

We can confirm that we have changed the group colors so that there is no overlap with previously used colors. We can also confirm that the p -value shown in this figure is the result of a single linear regression, where we model hemocrit as a function of rs679515 allele dosage (0,1, or 2), sex, race and age, as opposed to discrete comparison between genotype groups which would warrant a multiple test correction. We have updated the figure legend to better communicate this.

9. *The authors noted that microglia DA9 express CD83, supported by an image in Figure 5B showing microglia staining with CD83. However, Figure 5B does not conclusively demonstrate that cells expressing CD83 are exclusively associated with DA9, as this expression might not be unique to the DA9 subpopulation.*

We completed agree with the reviewer that Figure 5B (now **Figure 4B**) would not be sufficient to establish that CD83 is specific to the DA9 microglial subpopulation. Our intention in showing the CD83(+) microglia in **Figure 4B** was more of an illustration that what we had observed initially in the single nucleus RNAseq (CD83 as a highly over-expressed transcript in a subset of microglia) was also consistent with histochemical findings. Overall, we have found that the DA9 microglia are actually best characterized by multiple transcripts (**Supplementary Table 6**) of which CD83 was an exemplar, and also a convenient shorthand label since it has been described previously in the literature as a microglial marker. Within our snRNA-seq data set, we did observe some small number of microglia within DA9 that do not express (detectable) CD83 and some other microglia that express CD83, yet are not overall classified as a DA9 microglia. Overall, these observations seem consistent with a picture in which CD83 is a helpful marker gene, but not necessary or sufficient to classify DA9 by itself and is perhaps better conceptualized as a shorthand for a higher dimensional signature. We have adapted the text of our manuscript to better emphasize this.

10. *Regarding DA9 microglia, the observation of upregulated Disease-Associated Microglia (DAM) signatures prompts a request for the authors to provide DAM gene set scores across different microglial clusters.*

Thanks to the reviewer for this suggestion. Although we did identify multiple disease associated (DA) cell clusters for some of the more abundant cell types (particularly oligodendrocytes), we actually only identified a single differentially abundant microglial cluster (DA9) within our data, so we don't have additional DA microglia clusters to enrich against external DAM signatures. This summary can be seen in the dot plot below, which shows transcript abundance for a variety of key microglial subtype (including

DAM) signatures against all our described DA cluster types. We have included this dot plot in **Supplementary Figure 3C**.

11. Can the authors verify the accuracy of the information on line 323 (Figure 4A) to ensure alignment with the intended markings or annotations.

Thank you to the reviewer for alerting us of this error, which should refer instead to **Figure 5F**.

REVIEWERS' COMMENTS

Reviewer #1 (Remarks to the Author):

The authors have adequately addressed most comments by the reviewers. However, unfortunately, there are remaining issues that would need to be fixed. These are detailed below.

Major points:

- There are three paragraphs in the introduction section – one on AD, one on microglia and a summary of the findings. The study, however, does not focus on microglia. The introduction would better serve its purpose if it was about the rationale for this multi-scale, multi tissue study of AD (e.g. importance of peripheral blood biomarkers in AD diagnosis and monitoring disease progression, relevance of gut-brain axis and the microbiome in AD pathogenesis, etc).
- It is not clear why rs1111832832 and rs9429780 eQTL for CR1 in oligodendrocytes was not followed and why the authors pursued another SNP instead, that initially did not show up in their analysis. Did these two loci not have an AD-specific interaction in this dataset?
- Figure 4B – arrowhead is pointing to a cell, which might be amoeboid, but is not CD83+, next to it is a cell that is CD83+ but it looks ramified – overall a more detailed analysis on hundreds of cells from multiple donors would be needed to draw any conclusions regarding the relationship between CD83 expression and microglia morphology
- Line 230 – This is very confusing, especially as there were other microglia that were CD83+ but were not part of the DA9 cluster. Please use DA9 for clarity throughout the manuscript instead of CD83+ and CD83-.
- The authors argue against conflating CD83 positivity with DA9 membership, but then in the TF analysis part they do just that. To confirm the validity of NR4A2 as a TF regulating DA9 microglia phenotype, they should explore its relationship in the transcriptomic as well as in the ATAC-seq datasets to other marker genes of DA9, not just CD83.

Minor points:

- Figure 1 A – what RNA extraction did the authors perform
- Figure 1B – color bars need to be labeled with the measure they are displaying
- Figure 1C, Figure 2C, Figure 3D – tables are usually not part of figures
- figures and figure panels should be labeled in the order they are referenced in the text (e.g. Figure 3B and 3A)
- Figure 2C, Figure 3D, Figure 4E – tables are usually not part of figures
- What do the dots represent in Figure 4D? Generally, the figure legends need more info on the content of the figures.

Reviewer #2 (Remarks to the Author):

The authors have sufficiently addressed my concerns.

Reviewer #2 (Remarks on code availability):

Code looks fine, but may be difficult for folks to run without having all of the data organized exactly as the authors do.

Reviewer #3 (Remarks to the Author):

Thank you for all the insightful and thorough reviews.

Reviewer #3 (Remarks on code availability):

The code looks fine and has documentation

Point-by-Point Responses to Reviewer Comments

We would like to thank the reviewers for the further constructive comments and suggestions that were provided in the reviews of our previous submission.

Reviewer #1 (Remarks to the Author):

The authors have adequately addressed most comments by the reviewers. However, unfortunately, there are remaining issues that would need to be fixed. These are detailed below.

Major points:

- There are three paragraphs in the introduction section – one on AD, one on microglia and a summary of the findings. The study, however, does not focus on microglia. The introduction would better serve its purpose if it was about the rationale for this multi-scale, multi-tissue study of AD (e.g. importance of peripheral blood biomarkers in AD diagnosis and monitoring disease progression, relevance of gut-brain axis and the microbiome in AD pathogenesis, etc).

Thank you to the reviewer for this feedback. We have modified our introduction to de-emphasize the use of snRNA-seq as a means to characterize microglia (although we have kept some mention of this, as the CD83(+) microglia are a major vignette within our study). We have also introduced a paragraph to better frame the value of additional data layers (particularly multi-tissue) to contextualize findings:

“Despite the value of increasingly detailed molecular characterizations of brain tissue from subjects with AD, the development of a sophisticated understanding of the clinical and neuropathological context for identified cell subtypes and molecular networks is necessarily limited by the resolution of available antemortem and postmortem characterizations. Further, potentially informative cross-tissue interactions (e.g. gut-brain) are masked by a paucity of biorepositories that routinely collect brain and peripheral tissues from the same subjects. In addition to illuminating disease biology, multi-tissue profiling can offer valuable opportunities to identify peripheral biomarkers that might indicate disease-relevant brain states and treatment responses.”

- It is not clear why rs1111832832 and rs9429780 eQTL for CR1 in oligodendrocytes was not followed and why the authors pursued another SNP instead, that initially did not show up in their analysis. Did these two loci not have an AD-specific interaction in this dataset?

Thank you for the opportunity to clarify this thread of reasoning. We have updated the text to better reflect the motivation and justification for this:

“We observed that reported two AD risk loci, rs111183283³¹ (chr1:207677194) and rs9429780²⁵ (chr1: 1:207493845) are both eQTLs for Complement Receptor 1 (CR1) expression in oligodendrocytes (**Supplementary Table 4**). Recently, Fujita et al²⁸ and Mathys et al³² published analyses based on snRNA-seq from 433 and 427 dorsolateral prefrontal cortex samples respectively, from subjects included within the Religious Orders Study and Rush Memory and Aging Project (ROSMAP) cohort³³. Using these data (Fujita-ROSMAP and Mathys-ROSMAP DLPFC snRNA-seq) we confirmed the association between both loci and CR1 expression in oligodendrocytes (**Supplementary Table 4**). Fujita et al²⁸ also reported the nearby most significant AD risk locus rs679515 (chr1:207577223, Major allele: C, Minor allele: T, Reference allele: T) as an eQTL in oligodendrocytes, although this variant had not emerged in our analysis. Examination of our eQTL

results revealed that rs679515 had been removed following linkage disequilibrium-based pruning with rs9429780. We then confirmed that rs679515 was also an eQTL for CR1 in oligodendrocytes in our dataset by a direct association test, in agreement with Fujita et al²⁸ (Figure 3A, **Supplementary Table 4**), where minor allele T (also the AD risk allele) is associated with higher CR1 expression. Out of the three loci under consideration (rs11118328, rs9429780 and rs679515), rs679515 was most strongly associated with CR1 expression within oligodendrocytes. Consistent with Fujita et al²⁸, we also observed colocalization of the genetic signal driving CR1 expression in oligodendrocytes with AD GWAS risk signal³⁴ within the Banner cohort (posterior probability: 0.985), with rs679515 emerging as the most likely causal variant explaining this shared signal (posterior probability: 0.817, **Figure 3B, Supplementary Table 4**). Further examination of the association between these three loci and oligodendrocyte expression of CR1 within the Banner SFG data revealed that this was driven by the association within AD subjects (**Figure 3A, Supplementary Table 4**), indicating an AD-specific interaction for all three loci, but most strongly for rs679515 (AD diagnosis/dosage interaction pvalue = 2.1e-5). We did not observe a significant AD-specific interaction within the Fujita-ROSMAP or Mathys-ROSMAP data, (**Supplementary Figure 2A, Supplementary Table 5**). Whether the difference in detection of an AD-specific interaction between rs679515 and CR1 expression is explained by differences in cohort composition or technical variation remains to be investigated.”

- Figure 4B – arrowhead is pointing to a cell, which might be amoeboid, but is not CD83+, next to it is a cell that is CD83+ but it looks ramified – overall a more detailed analysis on hundreds of cells from multiple donors would be needed to draw any conclusions regarding the relationship between CD83 expression and microglia morphology

We agree with the reviewer and have removed reference to specific morphological differences between CD83(+) microglia and non CD83 microglia. We will perform a more systematic characterization of morphological properties of these cells in a forthcoming study.

- Line 230 – This is very confusing, especially as there were other microglia that were CD83+ but were not part of the DA9 cluster. Please use DA9 for clarity throughout the manuscript instead of CD83+ and CD83-.

We appreciate the reviewer’s perspective here. While it is true that CD83 is not the only transcript that characterizes these microglia, it is the most strongly differentially expressed, while also encoding a cell-surface expressed protein that has been reported as a microglial marker in the scientific literature. For this reason we’d respectfully propose leaving the language around CD83(+)/CD83(-) in place, with the caveat that these labels are helpful but imperfect. We have added some additional clarification in the text to highlight this:

“Herein we refer to DA9 microglia as CD83(+) microglia for brevity, however we note that these cells are characterized by several marker genes beyond CD83 and that our data includes other microglia that express CD83, but which are not members of the same DA subpopulation. Despite this, given the reports of CD83(+) microglia within the scientific literature already^{8,40}, we reasoned that CD83(+) would be an informative, though imperfect label. We detected CD83(+) microglia in 47% of AD (n=31 of 66) and 25% of Aged Control (n=9 of 35) subjects profiled within the Banner SFG snRNA-seq study.”

- The authors argue against conflating CD83 positivity with DA9 membership, but then in the TF analysis part they do just that. To confirm the validity of NR4A2 as a TF regulating DA9 microglia phenotype, they should explore its relationship in the transcriptomic as well as in the ATAC-seq datasets to other markers genes of DA9, not just CD83.

Thank you to the reviewer for highlighting this. We have revised our language to emphasize the manner in which this analysis identifies NR4A2 as a TF for CD83, rather than framing it as an inducer of CD83(+) / DA9 microglia per se. We propose that CD83 is worth this kind of focused attention since it was the most strongly differentially expressed gene within these microglia, as well as for its known biology as a marker of activated dendritic cells, which is consistent with a potential gut-microbial association that was eventually implicated by the finding of elevated immunoglobulin IgG4 within the transverse colon.

“To better understand the potential transcriptional context that may mediate the expression of CD83 in microglia, we constructed a transcriptional regulatory model for CD83 as learned from independent microglial transcriptomic data.”

Minor points:

- Figure 1 A – what RNA extraction did the authors perform*
- Figure 1B – color bars need to be labeled with the measure they are displaying*

Thank you, we have updated both of these items in a revised version of Figure 1.

- Figure 1C, Figure 2C, Figure 3D – tables are usually not part of figures*
- Figure 2C, Figure 3D, Figure 4E – tables are usually not part of figures*

We do concur with the reviewer that tables are not traditionally integrated into figures, however for this study where each main figure is aiming to progress a quite complex sequence of analyses and build upon successive findings, the inclusion of tables gives a convenient way to summarize certain aspects of those progressions, such as enrichment analysis results, summary statistics etc. Overall, we feel that the narrative value of each figure would be decreased if we were to separate out these current tables and then try to allude to their relevance only using the text.

- figures and figure panels should be labeled in the order they are referenced in the text (e.g. Figure 3B and 3A)

Thanks to the reviewer, we have corrected this previous mis-ordering and can confirm that Figure 3A is introduced prior to Figure 3B.

- What do the dots represent in Figure 4D? Generally, the figure legends need more info on the content of the figures.

Thanks to the reviewer for this opportunity to clarify. We can confirm that each value shown is a per-subject, averaged value for APOE expression, stratified by microglia type.

“and **(D)** at increased expression in CD83(+) microglia compared with CD83(-) microglia. Mean APOE expression for each subject (stratified by microglial type) shown.”

Reviewer #2 (Remarks to the Author):

The authors have sufficiently addressed my concerns.

Reviewer #2 (Remarks on code availability):

Code looks fine, but may be difficult for folks to run without having all of the data organized exactly as the authors do.

Our thanks to the reviewer for the thoughtful review and comments on our manuscript.

Reviewer #3 (Remarks to the Author):

Thank you for all the insightful and thorough reviews.

Reviewer #3 (Remarks on code availability):

The code looks fine and has documentation

Our thanks to the reviewer for the thoughtful review and comments on our manuscript.